# DNA damage induced by CDK4 and CDK6 blockade triggers anti-tumor immune responses through cGAS-STING pathway

Huimin Fan [1], Wancheng Liu[2], Yanqiong Zeng[1], Ying Zhou[3], Meiling Gao[1], Liping Yang[4], Hao Liu[1,5], Yueyue Shi[1], Lili Li[1], Jiayuan Ma[1], Jiayin Ruan[1,2], Ruyun Cao[1,5], Xiaoxia Jin [6✉], Jian Chen [6✉], Genhong Cheng [7✉] & Heng Yang [1,2✉]

CDK4/6 are important regulators of cell cycle and their inhibitors have been approved as anti-cancer drugs. Here, we report a STING-dependent anti-tumor immune mechanism responsible for tumor suppression by CDK4/6 blockade. Clinical datasets show that in human tissues, CDK4 and CDK6 are over-expressed and their expressions are negatively correlated with patients' overall survival and T cell infiltration. Deletion of *Cdk4* or *Cdk6* in tumor cells significantly reduce tumor growth. Mechanistically, we find that *Cdk4* or *Cdk6* deficiency contributes to an increased level of endogenous DNA damage, which triggers the cGAS-STING signaling pathway to activate type I interferon response. Knockout of *Sting* is sufficient to reverse and partially reverse the anti-tumor effect of *Cdk4* and *Cdk6* deficiency respectively. Therefore, our findings suggest that CDK4/6 inhibitors may enhance anti-tumor immunity through the STING-dependent type I interferon response.

[1] National Key Laboratory of Immunity and Inflammation, Suzhou Institute of Systems Medicine, Chinese Academy of Medical Sciences & Peking Union Medical College, Suzhou 215123, China. [2] Department of Hematology, Qilu Hospital of Shandong University, Jinan, Shandong 250012, China. [3] Department of Obstetrics and Gynecology, The First Affiliated Hospital of Soochow University, Suzhou 215006, China. [4] Department of Gastroenterology, Zhejiang Provincial People's Hospital, People's Hospital of Hangzhou Medical College, No. 158 Shangtang Road, Hangzhou, Zhejiang, China. [5] Department of Pharmacy, China Pharmaceutical University, No. 24 Tongjiaxiang Road, Nanjing 210009, China. [6] The Affiliated Tumor Hospital of Nantong University, Nantong Tumor Hospital, Nantong, Jiangsu, China. [7] Department of Microbiology, Immunology & Molecular Genetics, University of California, Los Angeles, CA, USA. ✉email: jxxntu@163.com; 3529995758@qq.com; gcheng@mednet.ucla.edu; yhmyt@hotmail.com

Cyclin-dependent kinases (CDKs) are a family of serine/threonine kinases that play a crucial role in modulating the stability of cell cycle-related proteins during cell cycle progression[1,2]. CDKs are activated upon interaction with their partner cyclins (A, B, D, and E)[3]. Among the cyclins, D-type cyclins (D1, D2 and D3) can interact with CDK4 and CDK6[4], a pair of cell cycle kinases that are similar in structure and function, mediating cell cycle transition from G0/G1-phase to S-phase[4,5]. Since uncontrolled cell proliferation is a hallmark of cancer, chemicals targeting CDKs to inhibit cell proliferation could be effective anti-tumor drugs[6,7]. Selective inhibitors of CDK4/6 including palbociclib, abemaciclib and ribociclib have been approved by US Food and Drug Administration (FDA) to treat postmenopausal women with breast cancer[8,9]. In recent years, researchers found that *Cdk4/6* inhibitors not only inhibit cell proliferation, but also trigger anti-tumor immune responses[10–12]. For example, Goel et al. reported that CDK4/6 inhibitor triggered type III interferon and thus activated anti-tumor immune responses[12]. CDK4/6 inhibitor was also reported to increase T cell activity by suppressing nuclear factor of activated T cell (NFAT) protein family and its target genes[10]. Meanwhile, CDK4 destabilized PD-L1 protein by modulating the phosphorylation of speckle type POZ protein (SPOP) to regulate cancer immune surveillance in multiple solid tumors[11]. Furthermore, CDK4/6 inhibitor suppressed regulatory T cell (Treg) proliferation and enhanced the activity of effector T cells[12]. These findings suggested that CDK4/6 may affect anti-tumor immunity. However, the underlying mechanisms of CDK4 and CDK6 in cancer cells involved in regulating anti-tumor responses remain to be elucidated.

Cyclic GMP-AMP synthase (cGAS) is a cytosolic DNA sensor that triggers immune responses against microbial pathogens such as DNA viruses[13]. Activation of cGAS stimulates the adapter protein stimulator of interferon genes (STING), which recruits TANK binding kinase-1 (TBK1), interferon regulatory factor 3 (IRF3) and finally triggers type I interferon (IFN) production[14–16]. In addition to its well known function in antiviral immunity, the cGAS-STING-IFN signaling pathway have recently shown to play important roles on cancer and other cellular functions such as DNA repair. DNA-damage induced by chemotherapy can be sensed by cGAS enzyme, resulting in the synthesis of the cyclic dinucleotide (cGAMP), which activates STING-dependent interferon induction pathway[17]. The interferon pathway plays a crucial role in anti-tumor responses by increasing infiltration and activation of immune cells, such as T cells, dendritic cells (DCs) and natural killer (NK) cells[18,19]. Furthermore, tumor-derived DNA can be transferred and released into the cytosol of macrophages and DCs[20], which in turn, activates cGAS-STING-IRF3-induced IFN signaling and enables DCs to present tumor-antigen and prime CD8+ T cells for anti-tumor immunity[21]. The crucial role of the cGAS-STING pathway as an activator of both innate and adaptive immune responses highlights the importance of DNA damaging therapies. Therapeutic agents including radiation, poly-(ADP-ribose) polymerase inhibitors (PARPi), and etoposide can generate cytosolic DNA and induce STING-dependent interferon production for anti-tumor immunity[22–24]. The correlation between DNA-damaging therapies and cGAS-STING pathway triggers more interest on combinatorial strategies to eliminate tumor cells through activation of immune responses.

Despite current studies on the effect of CDK4/6 inhibition on anti-tumor immunity, most of them used CDK4/6 inhibitors, which can cause off-target effect. For example, CDK4/6 inhibitors palbociclib and abemaciclib also have inhibitory activity on other CDKs such as CDK2[8]. So the mechanisms of CDK4/6 in anti-tumor immunity studied with CDK4/6 inhibitors lack accuracy.

The role of anti-tumor immunity triggered by CDK4/6 inhibitor has been previously reported by Goel et al.[12] through activation of IFN response by increased intracellular levels of endogenous retrovirus. In our previous study, we found that *Cdk2^{−/−}* cancer cells also had increased intracellular levels of endogenous retroviral double-stranded RNA and hyper-activation of interferon response through the MAVS-dependent signaling pathway[25]. Interestingly, recent studies showed that abemaciclib used by Goel et al. not only inhibited CDK4/6 but also inhibited CDK2. In this study, we used Crisper-Cas9 technology to generate *Cdk4* and *Cdk6* knockout cell lines to elucidate the role of CDK4 and CDK6 in anti-tumor immunity accurately.

In the current studies, we addressed the mechanisms responsible for the anti-tumor immunity of CDK4 and CDK6 blockage using *Cdk4^{−/−}* and *Cdk6^{−/−}* tumor cells. We have provided evidence that either *Cdk4* or *Cdk6* knockout triggers DNA damage of cancer cells and retards tumor growth in vivo through activation of STING-dependent type I interferon induction pathway and anti-tumor immune responses.

## Results

**CDK4 expression promotes tumor growth in an immune-dependent manner in vivo.** Although CDK4/6 inhibitors for breast cancer have achieved remarkable clinical success[12,26,27], the specific mechanisms of how CDK4 and CDK6 affect tumor micro-environment (TME) and anti-tumor immunity are still poorly understood. Recently, it is reported that aberrant expression of cancer genes can affect tumor development or regression by reshaping TME[28]. We aim to find the similarities and differences between CDK4 and CDK6 in regulating tumor immunogenicity. Analysis of The Cancer Genome Altas (TCGA) using the UALCAN database indicated that CDK4 mRNA expression was upregulated in lung adenocarcinoma (LUAD) samples and lung squamous cell carcinoma (LUSC) samples compared with normal samples (Supplementary Fig. 1a, b). Moreover, we found that lung cancer and sarcoma (SARC) patients with high CDK4 expression have lower overall survival (OS) rate than those with low CDK4 expression (Supplementary Fig. 1c, d) by searching the Kaplan-Meier Plotter database.

Next, we assessed the effect of palbociclib (an inhibitor of CDK4/6) on tumor cell growth in vitro. 0.5 μM Palbociclib was efficient to inhibit MCA205 cell growth at 72 h (Supplementary Fig. 1e). To evaluate the effect of palbociclib on cell cycle, flowmetry analysis of cell cycle was conducted. The percentage of cells in G1 phase was increased after treatment with palbociclib for 5 days and 10 days, indicating that palbociclib could cause G1/S arrest (Supplementary Fig. 1f, g). However, not all the cells hold a penerant G1 arrest as approximately 55% cells could still transit to S and G2/M phase. We carried on to examine the effect of palbociclib on tumor growth in vivo. Result showed that tumor growth of immunocompetent (C57BL/6 N) mice injected with palbociclib was significantly slower than those with PBS group (Fig. 1a), suggesting that palbociclib could effectively prevent subcutaneous tumor growth. Interestingly, only a slight difference in tumor growth was observed between palbociclib and PBS treated group when tumor implanted in nude mice, which are lack of mature T cells (Fig. 1b). These results indicated that the anti-tumor effect of palbociclib relied on immune system. To further clarify the specific mechanisms responsible for the regulation of tumor growth by aberrant CDK4 expression, we established *Cdk4*-deficient MCA205 mouse fibrosarcoma cell line and *Cdk4*-deficient TC1 mouse lung epithelial cell line by CRISPR/Cas9 technology with *Cdk4* specific sgRNA pairs. Control and *Cdk4*-deficient cancer cells were subcutaneously injected to immunocompetent mice. The results showed that

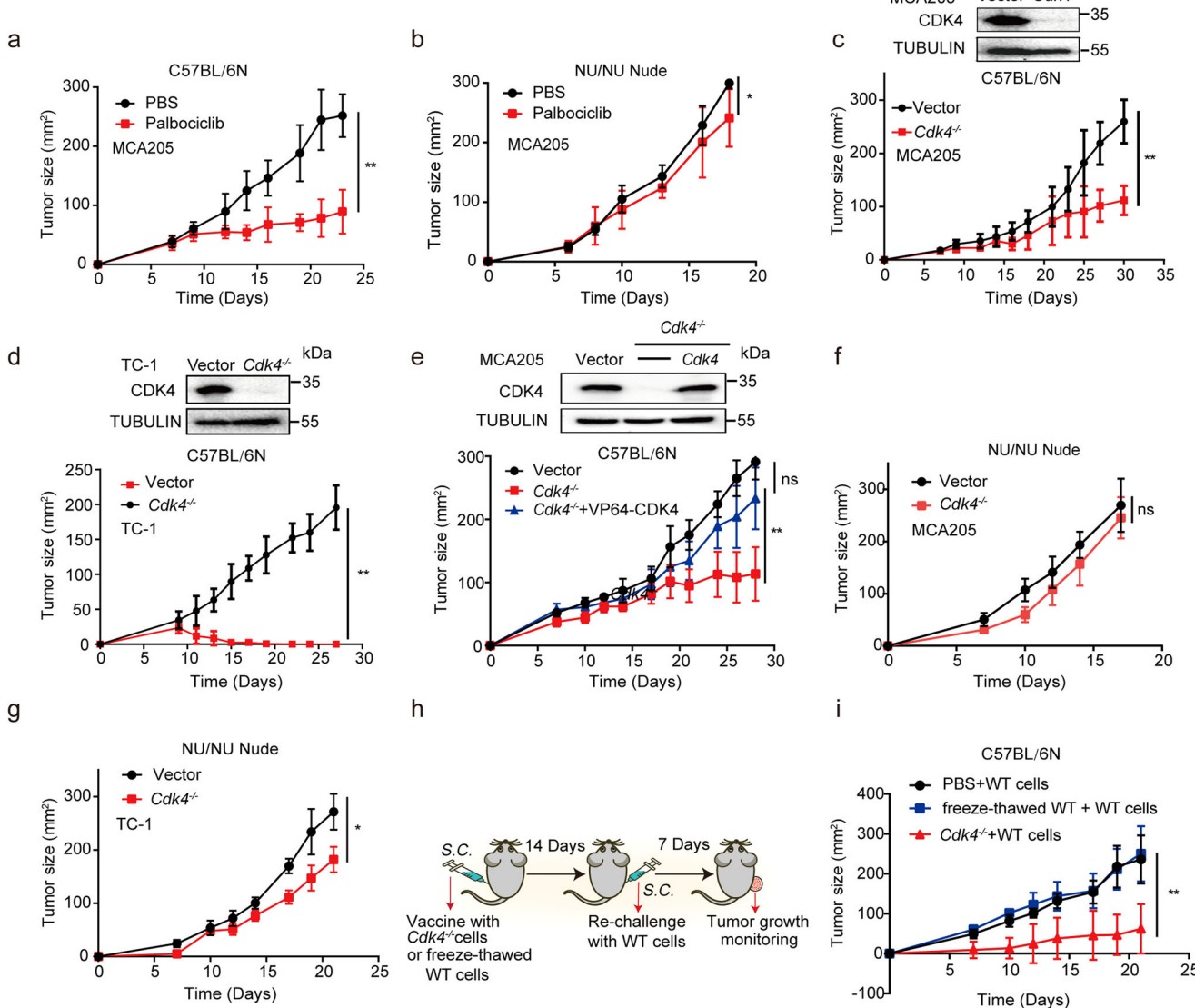

**Fig. 1 CDK4 expression enhances tumor growth in an immune-dependent manner in vivo. a** MCA205 tumor growth curve of C57BL/6 N mice with intratumoral injection of palbociclib (5 mM) or PBS. WT tumor cells ($2 \times 10^6$/mouse) were transplanted subcutaneously on the right flank of C57BL/6 N mice ($n = 5$) with palbociclib (5 mM) or PBS treatment at day 6, 7 and 8. **b** MCA205 tumor growth in nude mice with intratumoral injection of palbociclib (5 mM) or PBS. WT tumor cells ($2 \times 10^6$/mouse) were transplanted subcutaneously on the right flank of Nu/Nu mice ($n = 5$) with palbociclib (5 mM) or PBS treatment at day 6, 7 and 8. **c** Vector and $Cdk4^{-/-}$ MCA205 tumor growth curve in C57BL/6 N mice ($n = 5$). **d** Vector and $Cdk4^{-/-}$ TC-1 tumor growth curve of in C57BL/6 N mice ($n = 5$). **e** Tumor growth curve of vector, $Cdk4^{-/-}$ and $Cdk4$-restored MCA205 cells in C57BL/6 N mice ($n = 5$). **f** Vector and $Cdk4^{-/-}$ MCA205 tumor growth curve in nude mice ($n = 5$). **g** Vector and $Cdk4^{-/-}$ TC-1 tumor growth curve in nude mice ($n = 5$). **h**, **i** Tumor growth curve of C57BL/6 N mice re-challenged with WT MCA205. C57BL/6 N mice were first injected with PBS or $1 \times 10^6$ $Cdk4^{-/-}$ MCA205 cells or freeze-thawed WT cells (or PBS as control) on the left flank. The freeze-thaw cycles were conducted for 3 times. Mice ($n = 5$) were re-challenged with $2 \times 10^6$ WT MCA205 cells 14 days later on the right flank. WT MCA205 tumor size on the right flank was monitored. Data are representative of three independent experiments and presented as mean ± SD. Statistical significance was analyzed by the Mann–Whitney $U$ test. ns, no significant, *$p < 0.05$, **$p < 0.01$.

tumor bearing $Cdk4^{-/-}$ cells were much smaller than those bearing control cells (Fig. 1c, d). A $Cdk4$-restored cell line in $Cdk4^{-/-}$ MCA205 cells was also established to avoid off-target effects of the CRISPR/Cas9 system. $Cdk4$-restored cells, similar to control cells, developed tumors to larger sizes than $Cdk4^{-/-}$ cells (Fig. 1e). Interestingly, tumors grew to similar sizes in nude mice injected with either control or $Cdk4^{-/-}$ MCA205 and TC-1 cells (Fig. 1f, g), suggesting that the reduced tumor growth of $Cdk4$ deficient MCA205 and TC-1 cells was immune-dependent.

We further investigated whether animals immunized with $Cdk4^{-/-}$ cancer cells could form an immunological memory to protect host from tumor growth of WT cancer cells. C57BL/6 N mice were immunized with $Cdk4$ deficient cancer cells or freeze-thawed WT cells on the left flank, and two weeks later re-challenged with live WT cancer cells on the right flank (Fig. 1h). The results showed that immunization of mice with $Cdk4^{-/-}$ tumor cells could effectively suppress WT tumor growth on the right flank (Fig. 1i). These observations suggested that mice immunized with $Cdk4^{-/-}$ cancer cells could establish immunological memory, which effectively protected the host from re-challenging with corresponding WT cancer cells.

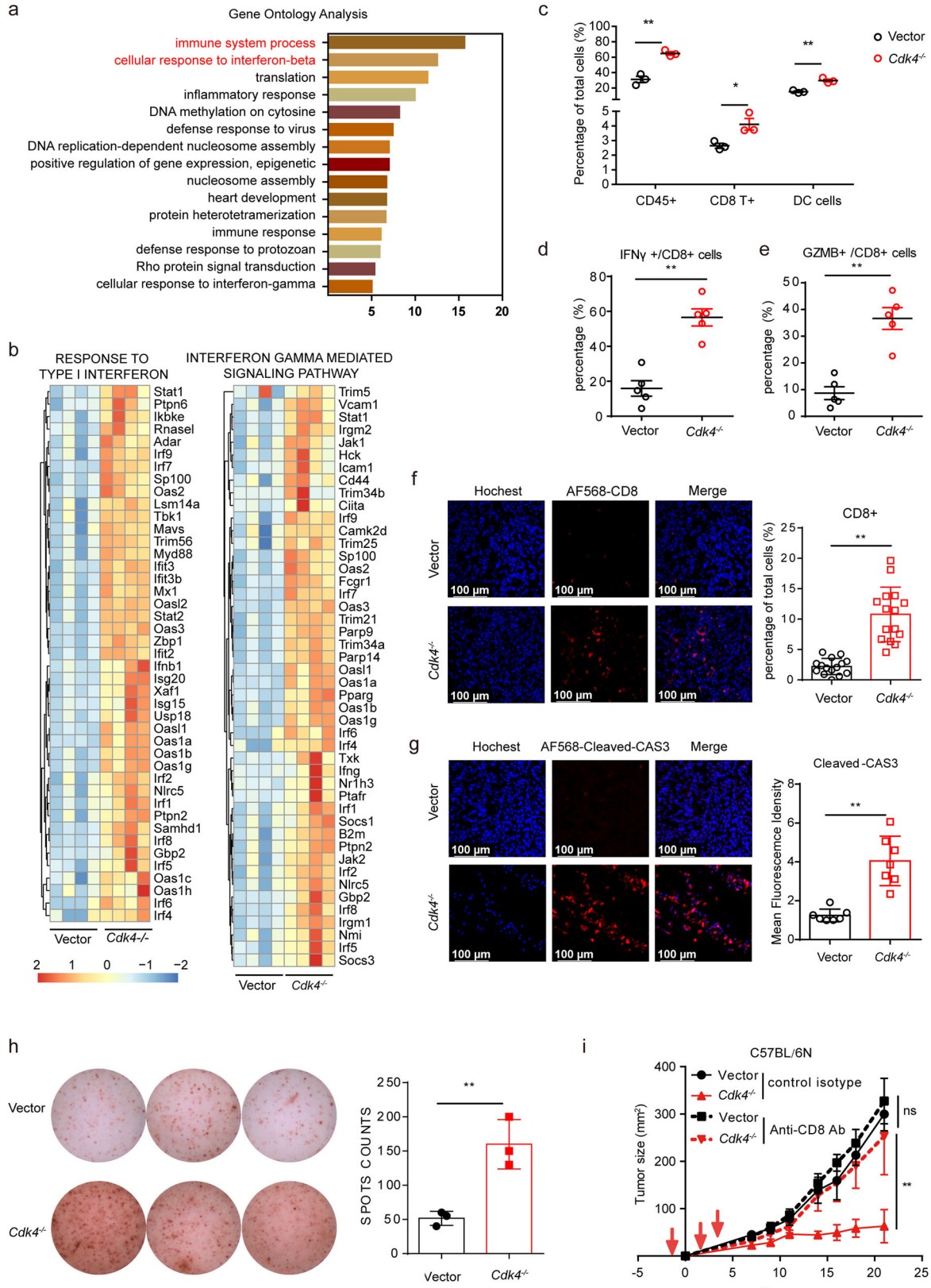

**Cdk4-deficiency triggers anti-tumor immune responses**. To better understand how *Cdk4* deletion in cancer cells delayed tumor growth in an immune dependent manner, transcriptome RNA sequencing (RNA-seq) was carried out using matched control and *Cdk4*$^{-/-}$ tumor tissues on day 10 after subcutaneous transplantation. Gene Ontology (GO) enrichment analysis of the

differentially expressed genes (DEGs) revealed numerous up-regulated genes related to immune responses, including "immune system process" and "cellular response to interferon-beta" (Fig. 2a). In addition, gene set enrichment analysis (GSEA) showed enrichment of signatures associated with "Type I interferon response" and "Interferon-g (IFN-γ) response" in *Cdk4*$^{-/-}$

**Fig. 2 Cdk4-deficiency triggers anti-tumor immune responses. a** Gene ontology analysis for significantly changed genes and related signaling pathway in MCA205 tumor tissues after *Cdk4* knockout. **b** GSEA analysis of signaling pathway changes of tumor tissues after *Cdk4* knockout and heat map of mRNA expression involved in "response to type I interferon" and "interferon gamma mediated signaling pathway" in vector and *Cdk4*$^{-/-}$ MCA205 tumor tissues. **c** Flow cytometry analysis of the infiltration of CD45.2$^+$ immune cells, CD45.2$^+$CD8$^+$ T cells, MHCII$^+$CD11c$^+$ DCs in vector or *Cdk4*$^{-/-}$ MCA205 tumor inoculated in C57BL/6 N mice. The percentage was quantified. **d, e** Flow cytometry analysis of the infiltration of CD8$^+$IFNγ$^+$ T cells (**d**), CD8$^+$GZMB$^+$ T cells (**e**) in vector or *Cdk4*$^{-/-}$ MCA205 tumor inoculated in C57BL/6 N mice. The percentage was quantified. **f** Immunofluorescence of vector or *Cdk4*$^{-/-}$ MCA205 tumor tissues stained with Hochest (blue, DNA) and antibodies against CD8 (AF568). Representative images are shown at 40× magnification. Scale bars, 100 μm. Each group includes at least five samples and visions were randomly chosen. **g** Immunofluorescence of vector or *Cdk4*$^{-/-}$ MCA205 tumor tissues stained with Hochest (blue, DNA) and antibodies against Cleaved-CASPASE3 (AF568). Representative images are shown at 40× magnification. Scale bars, 100 μm. Each group includes at least five samples and visions were randomly chosen. **h** IFN-γ secretion in vector and *Cdk4*$^{-/-}$ MCA205 tumor tissues was measured using ELISpot assay. **i** Tumor growth curve of C57BL/6 N mice xenografted vector or *Cdk4*$^{-/-}$ MCA205 with or without anti-CD8 antibody. Mice were intravenously injected with anti-CD8 antibody (200 μg/mouse) or control isotype on days −1, 3, and 5 as shown with red arrows. Vector or *Cdk4*$^{-/-}$ MCA205 tumor cells were subcutaneously transplanted on the right flank of C57BL/6 N mice on day 0. Data are representative of three independent experiments and presented as mean ± SD. Statistical significance was analyzed by the Mann–Whitney *U* test or unpaired Student's *t*-test. ns, no significant, *$p < 0.05$, **$p < 0.01$.

tumor tissues versus control tumor tissues (Fig. 2b). Flow cytometry analysis was also conducted to examine whether *Cdk4* deficient cancer cells could induce immune cell infiltration and activation in TME. Tumors were harvested on day 10 post implantation of either control or *Cdk4*$^{-/-}$ MCA205 cells to C57BL/6 N mice. Flow cytometry analysis revealed that there was an increased number of tumor-infiltrating immune cells including CD8$^+$ T cells and DCs in *Cdk4*$^{-/-}$ tumors compared with the control tumors (Fig. 2c and Supplementary Fig. 2a, b). Moreover, increased accumulation of T cells expressing effector molecules interferon-γ (IFN-γ) and Granzyme B (GZMB) was found in *Cdk4*$^{-/-}$ tumors compared with vector groups (Fig. 2d, e and Supplementary Fig. 2c). Furthermore, immunofluorescence also showed that more CD8$^+$ T cells were infiltrated in *Cdk4*$^{-/-}$ tumor tissues compared with vector group (Fig. 2f). In addition, more cleaved-caspase-3 positive cells were found in *Cdk4*$^{-/-}$ tumor tissues than control tumor tissues, indicating more tumor cells were undergoing apoptosis in *Cdk4*$^{-/-}$ tumors (Fig. 2g). IFN-γ ELISPOT assay was also performed to confirm the enhanced the production of IFN-γ in TME of *Cdk4*-deficient tumors (Fig. 2h).

In order to further explore whether the reduced tumor growth of *Cdk4*$^{-/-}$ cancer cells mainly relied on CD8$^+$ T cells, an anti-CD8 antibody was injected by intravenous to deplete CD8$^+$ T cells in C57BL/6 N mice. The tumor growth curve showed that anti-CD8 antibody was sufficient to increase tumor growth of *Cdk4*$^{-/-}$ similar to WT cancer cells (Fig. 2i). Collectively, these data indicated the increased infiltration and activation of CD8$^+$ T cells in tumor microenvironment might be responsible for the enhanced anti-tumor immunity associated with *Cdk4* deficient cancer cells.

**Cdk4-deficiency activates type I interferon pathway.** Although we have found that deletion of *Cdk4* in cancer cells increased immune cells infiltration and CD8$^+$ T cells activation in TME, its cellular mechanisms remained unclear. To further explore the underlying mechanisms, RNA-Seq analysis was performed using matched control and *Cdk4*$^{-/-}$ MCA205 cells to identify DEGs. Several significant GO categories were enriched ($p < 0.01$), including "cellular response to interferon-beta", "immune system process", and "innate immune response" (Fig. 3a). Through the GSEA pathway analysis, response to type I interferon pathways was also identified (Fig. 3b), and multiple interferon stmulatory genes (ISG), including *Stat1*, *Stat2*, *Isg15*, *Ifi204*, and *Ifit1* were up-regulated in *Cdk4*$^{-/-}$ MCA205 cells as compared to control MCA205 cells. We subsequently conducted RT-PCR to confirm the mRNA levels of these ISGs in *Cdk4*$^{-/-}$ or WT MCA205 cells treated with palbociclib. Results showed that both knockout of the

*Cdk4* gene and inhibition of the CDK4/6 activity could up-regulate these ISGs expression (Fig. 3c, d). Importantly, we detected the type I interferon expression in *Cdk4*$^{-/-}$ and control MCA205 cell culture supernatant using ISRE-L929 reporter cell lines. Results showed that type I interferon was released at higher levels in the supernatants of *Cdk4*$^{-/-}$ cells than control cells (Fig. 3e). Meanwhile, western blotting analysis confirmed that total and phosphorylated STAT1 were increased in *Cdk4*$^{-/-}$ MCA205 cells or WT cells treated with palbociclib (0.5 μM) compared with corresponding control cells (Fig. 3f, g). To confirm the hypothesis that *Cdk4* deficiency enhances anti-tumor immunity in vivo through the activation of type I interferon pathway, control and *Cdk4*$^{-/-}$ MCA205 cells were subcutaneously transplanted into *Ifnar1*$^{-/-}$ (Interferon-alpha/beta receptor) mice. As shown in Fig. 3h, all *Ifnar1*$^{-/-}$ mice inoculated vector or *Cdk4*$^{-/-}$ MCA205 cells developed tumor masses at similar rates, indicating that type I interferon played a crucial role in anti-tumor effect induced by *Cdk4*-deficiency.

**cGAS-STING pathway activated by DNA damage is required for Cdk4 medicated anti-tumor immunity.** The results above showed that the most significant GO category is "cellular response to interferon-beta". MAVS, an adaptor protein localized in the mitochondrial outer membrane, interacts with retinoic-inducible gene (RIG)-I-like receptors, which sense viral RNA molecules, while STING, an endoplasmic reticulum (ER)-resident adaptor protein, transduces DNA receptors mediated signals[29]. Both RIG-I-MAVS and cGAS-STING signaling pathways can stimulate type I IFN production.

In order to find out which gene is more critical in *Cdk4*$^{-/-}$ medicated anti-tumor responses, *Cdk4*$^{-/-}$/*Sting*$^{-/-}$ and *Cdk4*$^{-/-}$/*Mavs*$^{-/-}$ MCA205 cell lines were established by CRISPR/Cas9 technology with specific sg-RNA pairs (Fig. 4a, b). First, we observed that the growth of *Cdk4*$^{-/-}$ tumors was retarded comparing with control tumors as predicted, while *Cdk4*$^{-/-}$/*Sting*$^{-/-}$ tumors grew at faster rates similar as the control (Fig. 4c). In contrast, *Cdk4*$^{-/-}$/*Mavs*$^{-/-}$ tumor grew at slower rates similar as *Cdk4*$^{-/-}$ tumors (Fig. 4d). Protein levels of STAT1 and p-STAT1 upregulated in *Cdk4*$^{-/-}$ cells were restored in *Cdk4*$^{-/-}$/*Sting*$^{-/-}$ but not in *Cdk4*$^{-/-}$/*Mavs*$^{-/-}$ cells to the levels similar to WT MCA205 cells (Fig. 4e). Similarly, ISGs were up-regulated in *Cdk4*$^{-/-}$ and *Cdk4*$^{-/-}$/*Mavs*$^{-/-}$ cells, but not in *Cdk4*$^{-/-}$/*Sting*$^{-/-}$ cells as compared with control MCA205 cells (Fig. 4f, g). These findings suggested that the STING signaling pathway but not MAVS signaling pathway played an important role for the type I IFN up-regulation and enhanced anti-tumor activity of *Cdk4* knockout.

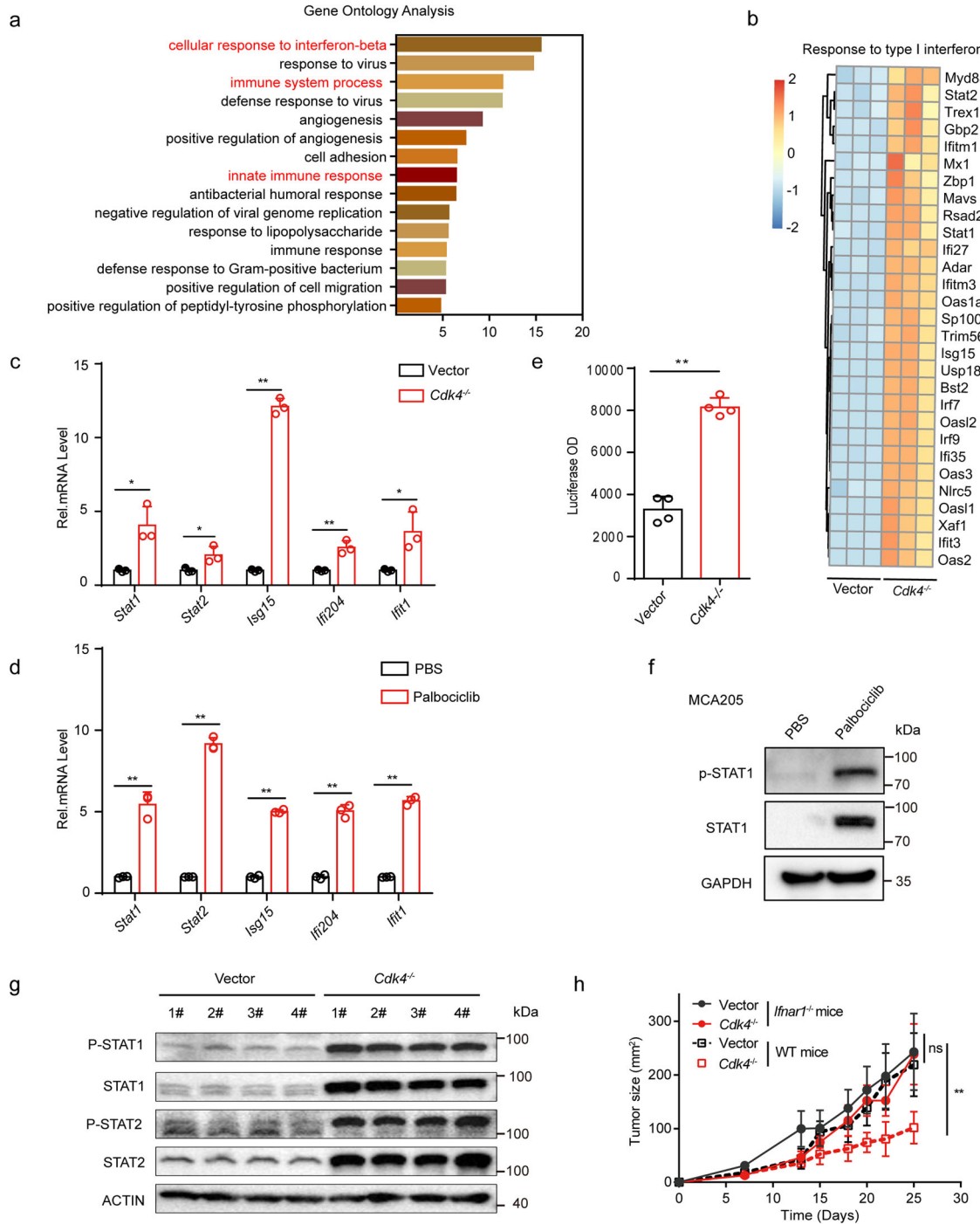

**Fig. 3 Cdk4-deficiency activates type I interferon pathway. a** Gene ontology analysis for significantly changed genes and related signaling pathway in *Cdk4* knockout MCA205 cells. **b** Heat map of mRNA expression involved in "response to type I interferon" in vector and *Cdk4*$^{-/-}$ MCA205 cells. **c** RT-PCR analysis of the expression of *Stat1, Stat2, Isg15, Ifi204, Ifit1* in vector and *Cdk4*$^{-/-}$ MCA205 cells. **d** RT-PCR analysis of the expression of *Stat1, Stat2, Isg15, Ifi204, Ifit1* in MCA205 cells treated with PBS or palbociclib (0.5 μM) for 10 days. **e** Luciferase OD value representing type1 IFN expression of vector and *Cdk4*$^{-/-}$ MCA205 detected by ISRE-L929 reporter cell line bioassay. **f** Protein expression of total and phosphorylated STAT1 in MCA205 cells treated with 0.5 μM palbociclib for 10 days. **g** Protein expression of total and phosphorylated STAT1 and STAT2 in Vector and *Cdk4*$^{-/-}$ MCA205 cells determined by Western Blot assay. **h** Vector and *Cdk4*$^{-/-}$ MCA205 tumor growth in WT and *Ifnar1*$^{-/-}$ mice. Data are representative of three independent experiments and presented as mean ± SD. Statistical significance was analyzed by the Mann–Whitney *U* test and unpaired Student's *t*-test. ns, no significant, *$p < 0.05$, **$p < 0.01$.

The cGAS/STING signaling pathway, triggered by cytosolic DNA, has been reported as a critical activator of anti-tumor immune responses[30]. Recent data indicated that the production of intracellular cytosolic DNA caused by DNA-damage can trigger innate immune responses[31]. We hypothesized that *Cdk4* deficiency might activate STING-dependent signaling caused by DNA-damage, leading to the activation of type I interferon pathway. Here, "Cytosolic DNA Sensing pathways" was identified in *Cdk4*$^{-/-}$ MCA205 cells by KEGG pathway analysis and multiple related genes were up-regulated in *Cdk4*$^{-/-}$ MCA205

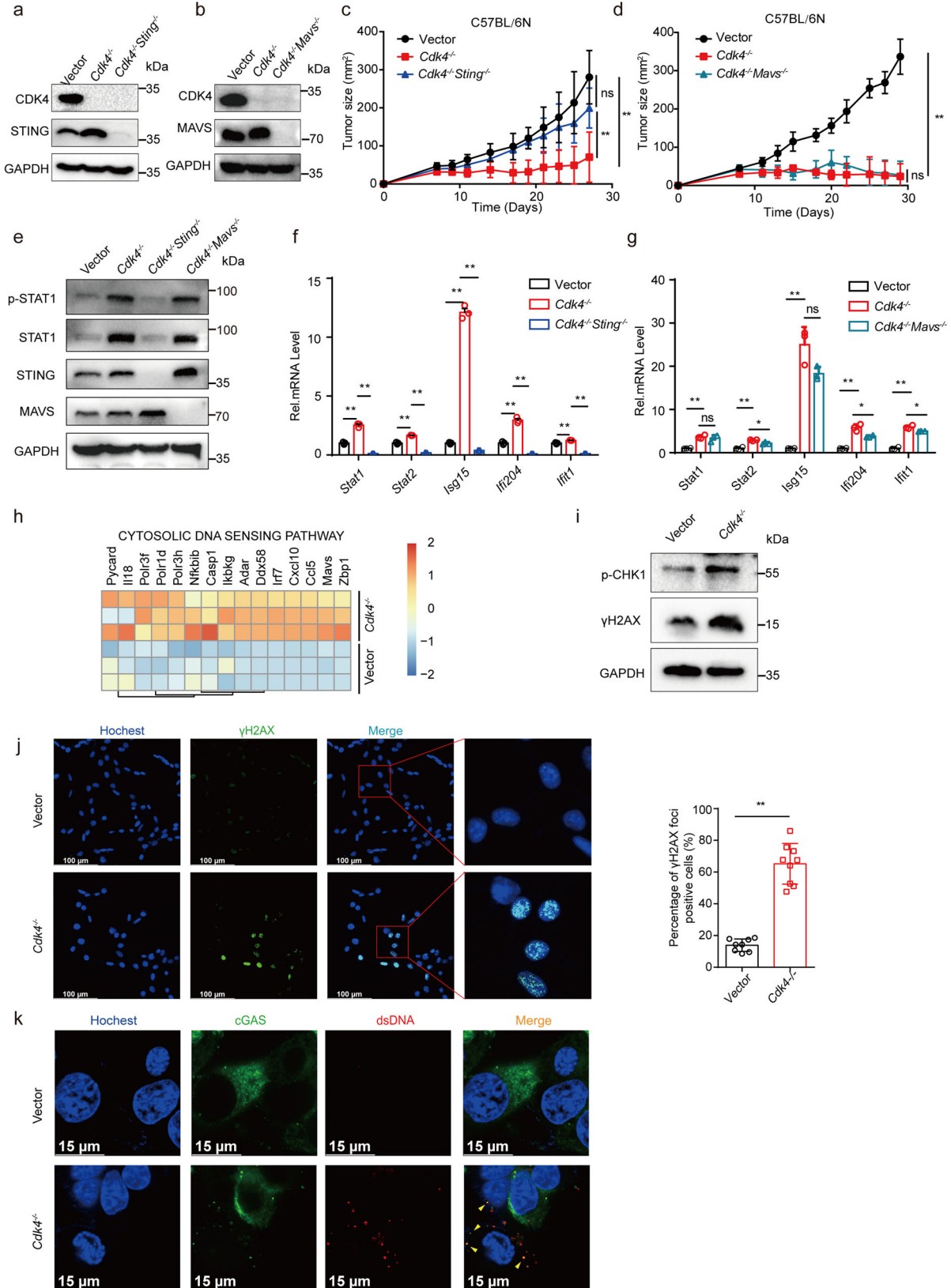

cells (Fig. 4h). Phosphorylated histone 2AX (γH2AX), which has been regarded as a general biomarker of DNA damage, is an acknowledged marker for DNA double-strand breaks (DSBs)[32]. Next, we performed western blot to detect some protein biomarkers for DNA damage including phospho-checkpoint kinase1 (p-CHK1) and p-H2AX (γH2AX). The results showed that $Cdk4^{-/-}$ tumor cells underwent more DNA damage than control cells (Fig. 4i). In addition, γH2AX foci formation was increased in $Cdk4^{-/-}$ cells compared with control cells (Fig. 4j). Importantly, cGAS co-localized with the enhanced cytosolic double-stranded DNA in $Cdk4^{-/-}$ cells (Fig. 4k), indicating that cytosolic dsDNA was sensed by cGAS and than activated STING

**Fig. 4 cGAS-STING pathway activated by DNA damage is required for *Cdk4* medicated anti-tumor immunity. a** Protein expression of CDK4 and STING in vector, *Cdk4*$^{-/-}$ and *Cdk4*$^{-/-}$*Sting*$^{-/-}$ dko MCA205 cells determined by Western Blot assay. **b** Protein expression of CDK4 and MAVS in vector, *Cdk4*$^{-/-}$ and *Cdk4*$^{-/-}$*Mavs*$^{-/-}$ dko MCA205 cells determined by Western Blot assay. **c** Vector, *Cdk4*$^{-/-}$ and *Cdk4*$^{-/-}$*Sting*$^{-/-}$ MCA205 tumor growth curve in C57BL/6 N mice. **d** Vector, *Cdk4*$^{-/-}$ and *Cdk4*$^{-/-}$*Mavs*$^{-/-}$ MCA205 tumor growth curve in C57BL/6 N mice. **e** Protein expression of total and phosphorylated STAT1 and STAT2 in vector, *Cdk4*$^{-/-}$, *Cdk4*$^{-/-}$*Sting*$^{-/-}$ and *Cdk4*$^{-/-}$*Mavs*$^{-/-}$MCA205 cells determined by Western Blot assay. **f** mRNA expression of *Stat1*, *Stat2*, *Isg15*, *Ifi204*, *Ifit1* by RT-PCR in vector, *Cdk4*$^{-/-}$ and *Cdk4*$^{-/-}$*Sting*$^{-/-}$ MCA205 cells. **g** mRNA expression of *Stat1*, *Stat2*, *Isg15*, *Ifi204*, *Ifit1* by RT-PCR in Vector, *Cdk4*$^{-/-}$ and *Cdk4*$^{-/-}$*Mavs*$^{-/-}$ MCA205 cells. **h** Heat map of relative mRNA expression associated with "cytosolic DNA sensing signaling pathway" in vector and *Cdk4*$^{-/-}$ MCA205 cells. **i** Phosphorylation-CHK1 and γH2AX protein expression determined by Western Blot assay in vector and *Cdk4*$^{-/-}$ MCA205 cells. **j** Immunofluorescence staining of vector and *Cdk4*$^{-/-}$ MCA205 cells stained with Hochest (blue, DNA) and antibodies against γH2AX (AF488). Representative images are shown at 60× magnification. Scale bars, 100 μm. Quantification of the percentage of γH2AX foci positive cells in total cells. **k** Immunofluorescence staining of vector and *Cdk4*$^{-/-}$ MCA205 cells stained with Hochest (blue, DNA), antibodies against dsDNA (AF568) and cGAS (AF488). Scale bars, 15 μm. Representative images are shown at 60× magnification. Data are representative of three independent experiments and presented as mean ± SD. Statistical significance was analyzed by the Mann–Whitney *U* test and unpaired Student's *t*-test. ns, no significant, *$p < 0.05$, **$p < 0.01$.

pathway. Together with previous results, we suggested that DNA damage caused by *Cdk4*-deficiency could activate cGAS-STING pathway, stimulate type I IFN and enhance anti-tumor immunity.

**_Cdk6_-deficiency inhibits tumor growth in vivo and vitro and activates type I interferon pathway.** As palbociclib inhibits both CDK4 and CDK6, we further addressed the effects of CDK6 on tumor growth and immunogenicity by analyzing online databases to determine whether CDK6 expression is related to the prognosis of patients. As shown in Supplementary Fig. 3a, b, the overall survival rate was negatively correlated with CDK6 expression in both lung caner and sarcoma. To further explore the function of CDK6 and the differences between CDK6 and CDK4, we established two independent *Cdk6*-deficient MCA205 mouse fibrosarcoma cell lines (A43 and A57) using CRISPR/Cas9 technology (Supplementary Fig. 3c). Control or the two *Cdk6*-deficient MCA205 clones were implanted to immunocompetent C57BL/6 N mice. The result showed that tumor growth of both two *Cdk6* deficiency cell lines were retarded as compared with those of control MCA205 cells (Fig. 5a). Since the two knockout cell lines showed the same phenotype, we chose clone A43 for later experiments. To investigate whether immune cells play an important role in *Cdk6*$^{-/-}$ induced tumor growth inhibition, a similar in vivo experiment was conducted in immune-deficiency nude mice. To our surprise, *Cdk6*-deficient tumors grew much slower than WT tumors in nude mice (Fig. 5b). As nude mice lack mature T cells but still have other immune cells, we investigated whether other immune cells such as NK might be involved in suppressing tumor growth.WT or *Cdk6*$^{-/-}$ tumor cells were subcutaneously implanted to NSG mice, which lack mature T cells, B cells and NK cells. Interestingly, *Cdk6*$^{-/-}$ tumors still grew slower and smaller than control tumors containing WT cells in NSG mice (Fig. 5c). To validate that these results were not cell line specific, we also generated two independent *Cdk6*$^{-/-}$ TC-1 cell lines clone 7# and 19# with Crisper-Cas9 technology (Fig. 3d). Similarly, *Cdk6* deficiency in TC1 could suppress tumor growth in both immunocompetent mice and immunodeficient mice (Fig. 5d, e).

Based on the tumor growth curve in nude and NSG mice, we speculated that *Cdk6* deficiency may directly affect cancer cell proliferation independent of immune responses. It is reported that heterozygous RB1 loss are biomarkers for CDK4/6 inhibitor resistance[33], so it was important to evaluate RB (which is phosphorylated by CDK4/6 to drive cell cycle process[8]) status of these two cell lines. As shown in Supplementary Fig. 3e, RB was expressed in WT, *Cdk4*$^{-/-}$ and *Cdk6*$^{-/-}$ cells and p-RB was decreased in *Cdk4*$^{-/-}$ and *Cdk6*$^{-/-}$ cells compared with WT cells, demonstrating that RB is functional in MCA205 cells. In order to compare the proliferative capacity of these three cell

types, cell counting kit-8 (CCK8) assay which was applied to detect cell proliferation and colony formation assay were conducted. As shown in Fig. 5f, g, *Cdk6*$^{-/-}$ MCA205 cells proliferated at a slower rate and formed fewer colonies than WT and *Cdk4*$^{-/-}$ MCA205 cells, indicating *Cdk6*-deficiency could directly affect cancer cell proliferation in vitro. Similar result were observed in *Cdk4*$^{-/-}$ and *Cdk6*$^{-/-}$ TC1 cells (Fig. 5h). Next we detected the cell cycle process in WT, *Cdk4*$^{-/-}$ and *Cdk6*$^{-/-}$ MCA205 cells. It was shown that the percentage of G1 phase cells were increased in both *Cdk4*$^{-/-}$ and *Cdk6*$^{-/-}$ cells compared with WT cells, demonstrating that *Cdk4* or *Cdk6* knockout could cause G1/S arrest (Supplementary Fig. 3f, g). It is reported that CDK4/6 inhibitors could induce senescence which promotes a senescence associated secretory phenotype (SASP)[34]. We further detected the SASP response of WT, *Cdk4*$^{-/-}$ and *Cdk6*$^{-/-}$ MCA205 cells. As *Cdk6*$^{-/-}$ cells, but not *Cdk4*$^{-/-}$ cells exhibited up-regulated p21 protein (Supplementary Fig. 3h) and senescence associated cytokines such as *Il-1α*, *Il-6*, *Cxcl1*, *Ccl2*, *Ccl7*, *Ccl8* and *Ccl20* (Supplementary Fig. 3i), it was interesting that unlike *Cdk4*$^{-/-}$ cells, *Cdk6*$^{-/-}$ cells were becoming senescent, which may explain why *Cdk6*$^{-/-}$ cells proliferate more slowly.

To determine if *Cdk6* deficient cells, similar to *Cdk4* deficient cells, have hyper-type I IFN response, we performed RNA-Seq analysis using matched vector and *Cdk6*$^{-/-}$ MCA205 cells to identify DEGs. GSEA pathway analysis identified upregulation of numerous ISGs in *Cdk6*$^{-/-}$ MCA205 as compared with WT MCA205 (Fig. 5i), part of which including *Stat1*, *Stat2*, *Isg15*, *Ifi204*, and *Ifit1* were confirmed by RT-PCR assay (Fig. 5j). ELISA assay was conducted to test the amount of IFNβ released in cell culture supernatants. As predicted, the release of IFNβ was increased in both *Cdk4*$^{-/-}$ and *Cdk6*$^{-/-}$ MCA205 cells (Fig. 5k). Furthermore, Western blot analysis also showed that the expression of STAT1 and p-STAT1 was higher in both two *Cdk6*$^{-/-}$ MCA205 clones than those in control cells (Fig. 5l).

To further evaluate the role of type I IFN pathway in tumor growth of *Cdk6*-deficient cancer cells, control or *Cdk6*$^{-/-}$ MCA205 cells were subcutaneously transplanted to *Ifnar1*$^{-/-}$ mice. As shown in Fig. 5m, *Cdk6*$^{-/-}$ cancer cells grew up to 100mm$^2$ in *Ifnar1*$^{-/-}$ mice, though still smaller than control group, were still larger than those in WT C57 mice. Collectively, the above results proved that *Cdk6*-deficiency could trigger type I interferon signaling pathway, which was partially required for tumor inhibition in vivo.

**cGAS-STING pathway activated by DNA damage is partially responsible for *Cdk6* medicated anti-tumor effect.** As mentioned above, RIG1-MAVS and cGAS-STING are the two classic pathways to trigger type I interferon signaling. We next sought to determine which pathway was more important in *Cdk6*$^{-/-}$ mediated

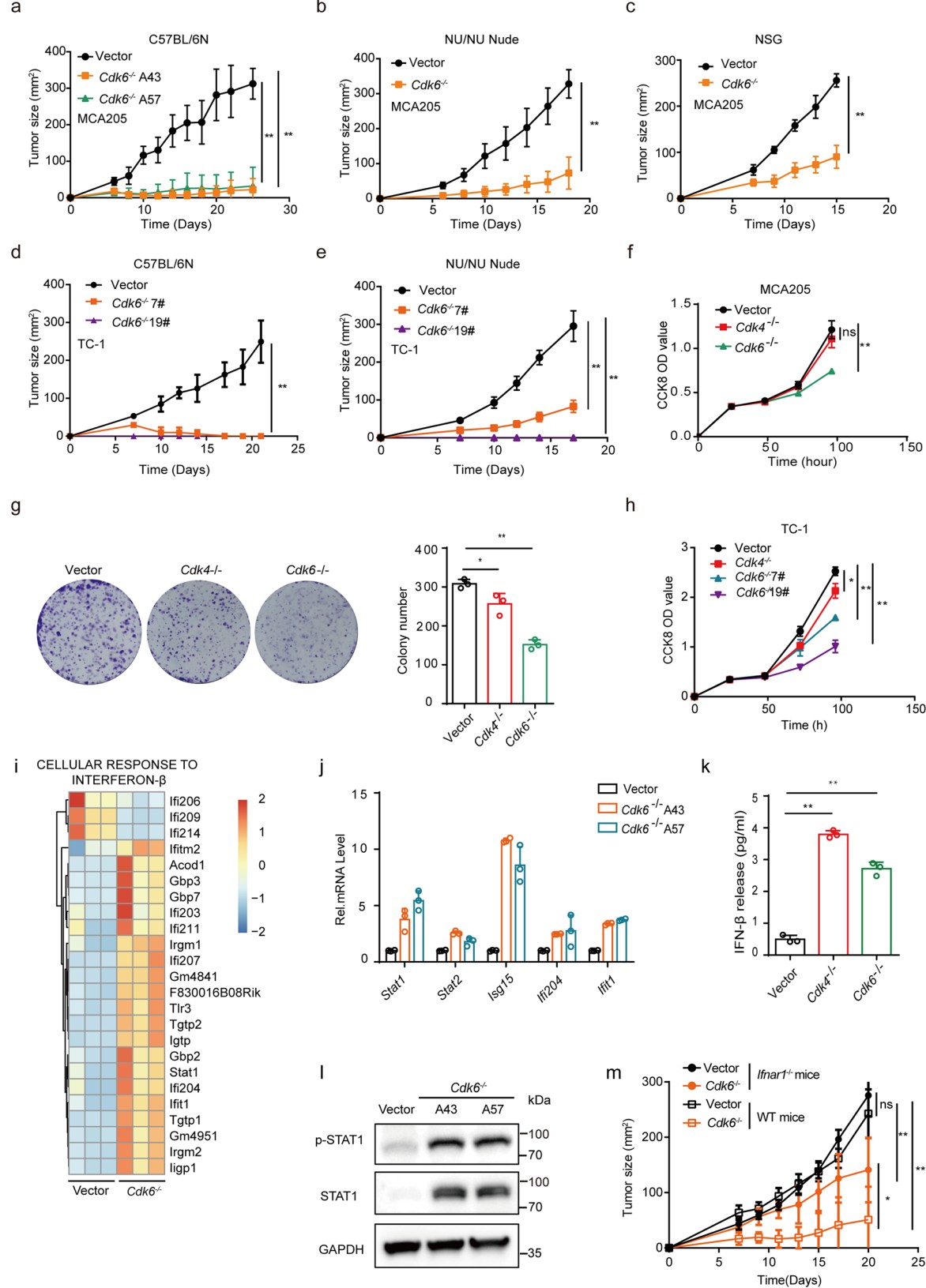

anti-tumor responses. To achieve this, $Cdk6^{-/-}/Sting^{-/-}$ and $Cdk6^{-/-}/Mavs^{-/-}$ MCA205 cell lines were established using CRISPR/Cas9 technology with specific sg-RNA pairs (Fig. 6a, b). As expected from prior experiments, the protein expression of STAT1 and p-STAT1 increased in $Cdk6^{-/-}$ cells. Though we saw a decrease of protein level of STAT1 and p-STAT1 in both double knockout cell lines compared with $Cdk6^{-/-}$ cells, $Cdk6^{-/-}/Sting^{-/-}$ cells showed more significant difference with $Cdk6^{-/-}$ cells (Fig. 6c). Similar result was shown in RT-PCR assay, that mRNA expression of *Stat1*, *Stat2*, *Isg15*, *Ifi204*, and *Ifit1* were up-regulated in $Cdk6^{-/-}$ cells. Both $Cdk6^{-/-}/Sting^{-/-}$ and $Cdk6^{-/-}/Mavs^{-/-}$ cell lines showed a decline of these genes compared with $Cdk6^{-/-}$ cells, but

**Fig. 5 *Cdk6*-deficiency inhibits tumor growth in vivo and vitro and activates type I interferon pathway. a** Vector and *Cdk6*$^{-/-}$ MCA205 tumor growth curve in C57BL/6 N mice. Vector or two clones of *Cdk6*$^{-/-}$ (A43 and A57) MCA205 tumor cells were transplanted subcutaneously on the right flank of C57BL/6 N mice (*n* = 5). **b** Vector and *Cdk6*$^{-/-}$ MCA205 tumor growth curve in Nu/Nu mice. **c** Tumor growth curve of vector and *Cdk6*$^{-/-}$ MCA205 in NSG mice. **d** Vector and *Cdk6*$^{-/-}$ TC1 tumor growth curve in C57BL/6 N mice. Vector or two clones of *Cdk6*$^{-/-}$ (7# and 19#) TC1 tumor cells were transplanted subcutaneously on the right flank of C57BL/6 N mice (*n* = 5). **e** Vector and *Cdk6*$^{-/-}$ TC1 (7# and 19#) tumor growth curve in Nu/Nu mice (*n* = 5). **f** Cell proliferation of vector, *Cdk4*$^{-/-}$ and *Cdk6*$^{-/-}$ MCA205 cells detected by CCK8 assay. **g** Colony formation assay conducted with vector, *Cdk4*$^{-/-}$ and *Cdk6*$^{-/-}$ MCA205 cells. **h** Cell proliferation of vector, *Cdk4*$^{-/-}$ and *Cdk6*$^{-/-}$ TC1 cells detected by CCK8 assay. **i** Heat map of mRNA expression involved in "cellular response to interferon-β" in vector and *Cdk6*$^{-/-}$ MCA205 cells. **j** mRNA expression of *Stat1*, *Stat2*, *Isg15*, *Ifi204*, *Ifit1* by RT-PCR in vector and two clones of *Cdk6*$^{-/-}$ (A43 and A57) MCA205 cells. **k** IFN-β expression in vector, *Cdk4*$^{-/-}$ and *Cdk6*$^{-/-}$ MCA205 cells supernatant detected by ELISA assay. **l** Protein expression of total and phosphorylated STAT1 in vector, *Cdk6*$^{-/-}$ (A43 and A57) MCA205 cells determined by Western Blot assay. **m** Vector and *Cdk6*$^{-/-}$ MCA205 tumor growth in *Ifnar1*$^{-/-}$ mice. Data are representative of three independent experiments and presented as and mean ± SD. Statistical significance was analyzed by the Mann–Whitney *U* test and unpaired Student's *t*-test. ns, no significant,*$p < 0.05$, \*\*$p < 0.01$.

*Cdk6*$^{-/-}$/*Sting*$^{-/-}$ cells presented more significant decrease (Fig. 6d). Finally, we subcutaneously implanted control, *Cdk6*$^{-/-}$, *Cdk6*$^{-/-}$/*Sting*$^{-/-}$ and *Cdk6*$^{-/-}$/*Mavs*$^{-/-}$ tumor cells to C57 mice and measured tumor size. Consistent with prior results, *Cdk6* deficiency resulted in sharp tumor regression. There was no significant difference between *Cdk6*$^{-/-}$/*Mavs*$^{-/-}$ and *Cdk6*$^{-/-}$ tumor growth, while *Cdk6*$^{-/-}$/*Sting*$^{-/-}$ tumors grew larger than *Cdk6*$^{-/-}$ tumors (Fig. 6e), suggesting that STING pathway was involved in *Cdk6* deficiency mediated anti-tumor effect.

Next, we assessed whether *Cdk6*-deficiency could also trigger intrinsic DNA damage. Immunofluorescence staining indicated increased γH2AX foci (Fig. 6f) and cytosolic dsDNA, which was co-localized with cGAS (Fig. 6g), were expressed in *Cdk6*$^{-/-}$ cells compared with control cells. Western blot assay confirmed that DNA damage markers p-CHK1 and γH2AX expression were upregulated in *Cdk6*$^{-/-}$ cells (Fig. 6h). Importantly, CDK4/6 inhibitor palbociclib could also induce cytosolic dsDNA and its co-localization with cGAS (Fig. 6i). Collectively, these results demonstrated that both *Cdk4*- and *Cdk6*-deficieny could trigger DNA damage. We further evaluated the cGAS-STING activation downstream of DNA damage in WT, *Cdk4*$^{-/-}$ and *Cdk6*$^{-/-}$ MCA205 cells. As shown in Fig. 7a, the expression of cGAS, p-TBK1, p-STING and p-IRF3 were up-regulated in both *Cdk4*$^{-/-}$ and *Cdk6*$^{-/-}$ cells, proving the activation of cGAS-STING pathway in these two cell lines. In addition, MCA205 and 4T1 (breast cancer cell line) treated with palbociclib could also trigger cGAS-STING and downstream type I interferon pathway activation (Fig. 7b, c).

Due to CDK4/6 plays an important role in cell cycle, we speculated that *Cdk4* or *Cdk6* knockout may block DNA replication process. To verify this, we checked the RNA-seq data of *Cdk4*$^{-/-}$ cells and *Cdk6*$^{-/-}$ cells compared with WT cells. During DNA reproduction, MCM2-7 complex is responsible for unwinding DNA to allow replication fork progress[35] and DNA polymerase catalyzes the simultaneous replication of two strands of DNA. The RNA-seq results showed that genes involved in DNA replication including those encoding pre-initiation complex (five members of MCM2-7) and DNA polymerase subunits were down-regulated in *Cdk4*$^{-/-}$ cells (Fig. 7d). Interestingly, in *Cdk6*$^{-/-}$ cells, genes encoding pre-initiation complex were upregulated while genes encoding DNA polymerase subunits were decreased (Fig. 7d). This result was validated by QPCR assay, which showed a decrease expression of MCM family members (*Mcm2*, *Mcm3*, *Mcm4* and *Mcm5*) in *Cdk4*$^{-/-}$ cells and a decrease of DNA polymerase subunits (*Polg2* and *Poln*) in in *Cdk6*$^{-/-}$ cells (Fig. 7e). These results demonstrated that *Cdk4* deficiency affected DNA reproduction process including replication fork progress and DNA polymerase reaction while *Cdk6* deficiency only affect the DNA polymerase reaction. Taken together, our results showed that both *Cdk4* and *Cdk6* deficiency could cause DNA replication

stress, resulting in the activation of cGAS-STING pathway triggered by cytosolic DNA damage.

**CDK4 and CDK6 are highly expressed in breast cancers and ISGs were upregulated in patients treated with palbociclib.** In order to investigate the potential of CDK4 and CDK6 as biomarker for breast cancers, we collected histological samples of breast cancer patients (*n* = 125) including paired cancerous tissues and benign tissues (patients information seen in Table 1). Figure 8a–c demonstrated high expression of CDK4 and CDK6 in cancerous tissues compared with benign tissues. We also collected PBMC from breast cancer patients (*n* = 10) with or without CDK4/6 inhibitor treatment to evaluate ISG expression (patients information seen in Table 2). GSEA analysis revealed that multiple ISGs were up-regulated in PBMC of patients receiving palbociclib treatment (Fig. 8d).

Overall, our clinical and laboratory studies identified the similar but distinct role of CDK4 and CDK6 in anti-tumor responses. While targeting CDK6 can directly inhibit tumor cell proliferation, we put forward a pathway by which blocking CDK4 and CDK6 in cancer cells enhance anti-tumor immune responses by inducing DNA damage-mediated, STING-dependent type I IFN production (Fig. 8e). These findings may have guiding significance in clinical medication of CDK4/6 inhibitors.

## Discussion
Multiple CDK4/6 inhibitors such as palbociclib, abemaciclib, and ribociclib have been approved as anticancer drugs. Despite the central role of CDK4/6 in cell cycle G1/S transition, studies have shown that non-haematological cells without *Cdk4* and *Cdk6* are able to proliferate normally[36]. Interestingly, embryonic fibroblasts without *Cdk4* and *Cdk6* could also enter S phase, although with reduced efficiency, with cyclin Ds interacting with CDK2 to drive cell-cycle transition[36].

Since the primary function of CDK4/6, which is activated by the D-type cyclins D1, D2, and D3 (cyclin D), is to drive cell-cycle progression from G1 to S phase, it is believed that the major mechanism responsible for anti-tumor activity of CDK4/6 inhibitors is induction of cell cycle arrest. However, recent studies suggest that CDK4/6 inhibitors can also inhibit tumor growth through additional mechanisms including modulation of mitogenic kinase signaling, induction of a senescence-like phenotype[34], and enhancement of cancer cell immunogenicity[37]. In this report, we have identified a DNA damage-mediated cGAS-STING dependent type I IFN pathway activated in CDK4- and CDK6-deficient cells. More importantly, we have provided evidence that this CDK4/6-deficient triggered pathway plays an important role in enhancing anti-tumor immunity by CDK4/6 inhibitors. To study the role of CDK4 and CDK6 in tumor cells, it is important to evaluate Rb status as loss of RB leads to resistance of CDK4/6

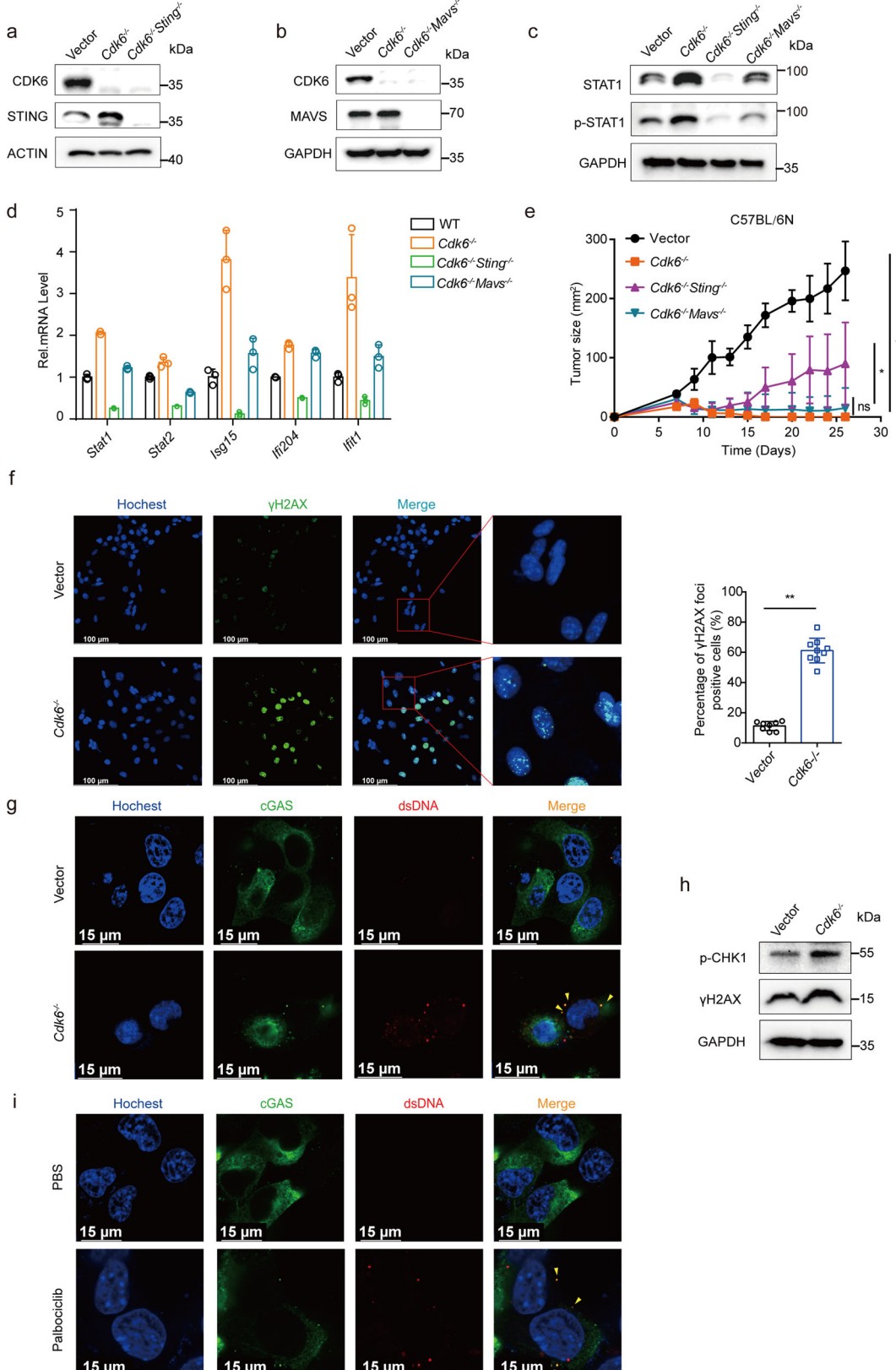

inhibitors[33,38]. We detected that RB was expressed in WT cells and the phosphorylation of RB was decreased in *Cdk4* and *Cdk6* knockout cells, demonstrating that RB1 is functional in MCA205. TC1 cells were obtained from C57BL/6 primary mouse lung cells transfected with HPV E6/E7 oncogenes[39]. E7 interacts with, and then inactivates RB, leading to uncontrolled cell proliferation[40]. So

our study applies to tumors both with functional RB and defective RB.

Recent studies have found that targeted anticancer agents can also regulate immune cells infiltration and activation in tumor microenvironment[41]. However, the main immunomodulatory effects of targeted anticancer agents and the specific regulatory

**Fig. 6 cGAS-STING pathway activated by DNA damage is partially responsible for *Cdk6* medicated anti-tumor effect. a** Protein expression of CDK6 and STING in vector, *Cdk6$^{-/-}$* and *Cdk6$^{-/-}$Sting$^{-/-}$* dko MCA205 cells determined by Western blot assay. **b** Protein expression of CDK6 and MAVS in vector, *Cdk6$^{-/-}$* and *Cdk6$^{-/-}$Mavs$^{-/-}$* dko MCA205 cells determined by Western blot assay. **c** Protein expression of total and phosphorylated STAT1 in vector, *Cdk6$^{-/-}$*, *Cdk6$^{-/-}$Sting$^{-/-}$* and *Cdk6$^{-/-}$Mavs$^{-/-}$* MCA205 cells determined by Western blot assay. **d** mRNA expression of *Stat1, Stat2, Isg15, Ifi204, Ifit1* by RT-PCR in vector, *Cdk6$^{-/-}$*, *Cdk6$^{-/-}$Sting$^{-/-}$* and *Cdk6$^{-/-}$Mavs$^{-/-}$* MCA205 cells. **e** Vector, *Cdk6$^{-/-}$*, *Cdk6$^{-/-}$Sting$^{-/-}$* and *Cdk6$^{-/-}$Mavs$^{-/-}$* MCA205 tumor growth curve in C57BL/6 N mice. **f** Immunofluorescence staining of vector and *Cdk6$^{-/-}$* MCA205 cells stained with Hochest (blue, DNA) and antibodies against γH2AX (AF488). Representative images are shown at 60× magnification. Scale bars, 100 μm. Quantification of the percentage of γH2AX foci positive cells in total cells. **g** Immunofluorescence staining of vector and *Cdk6$^{-/-}$* MCA205 cells stained with Hochest (blue, DNA), antibodies against dsDNA (AF568) and cGAS (AF488). Scale bars, 15 μm. **h** Protein expression of phosphorylation-CHK1 and γH2AX expression in vector and *Cdk6$^{-/-}$* MCA205 cells determined by Western blot assay. **i** Immunofluorescence staining of MCA205 cells treated with PBS or 0.5 μM Palbociclib for 10 days stained with Hochest (blue, DNA), antibodies against dsDNA (AF568) and cGAS (AF488). Scale bars, 15 μm. Data are representative of three independent experiments and presented as mean ± SD. Statistical significance was analyzed by the Mann–Whitney *U* test and unpaired Student's *t*-test. ns, no significant, *$p < 0.05$, **$p < 0.01$.

mechanisms need to be further explored. Tumors can be divided into "cold tumor" and "hot tumor" according to the immune cell infiltration in microenvironment, which plays an important role in controlling tumor growth. Here, we found that pharmacologic inhibition of CDK4/6 or deletion of CDK4 in tumor cells inhibited tumor growth only in the immunocompetent mice but not in the immunodeficient mice (Fig. 1). However, CDK6 deletion in tumor cells resulted in significant tumor retard in both immunocompetent mice and immunodeficient mice (Fig. 5). These results identified the crucial role of CDK4 in regulating tumor growth in an immune-dependent way. Flow cytometry and immunofluorescence staining showed that tumors developed from *Cdk4$^{-/-}$* cancer cells had increased DCs and CD8$^+$ T cells infiltration (Fig. 2c, f). Moreover, there were more infiltration of effector T cells marked as CD8$^+$IFN-γ$^+$ and CD8$^+$GZMB$^+$ in *Cdk4* deficiency tumor tissues as compared with tumors derived from the corresponding parantal cancer cells (Fig. 2d, e). CD8$^+$ T cell depletion could totally reverse the anti-tumor effect of *Cdk4* knockout, demonstrating the crucial role of CD8$^+$ T cells in anti-tumor immunity observed in *Cdk4* deficient tumors (Fig. 2i). RNA-seq analysis of tumor tissues showed increased expression of type I IFNs response genes in tumors developed from *Cdk4$^{-/-}$* cancer cells. Analysis of cell RNA-seq data demonstrated upregulation of multiple type I IFN-induced genes in *Cdk4* or *Cdk6* deficient cells, which were further validated by RT-PCR and western blotting analyses. Type I IFN signaling can enhance the capacity of DCs to cross-prime antigens, which is required for CD8$^+$ T cells to acquire an activated phenotype and tumor cell-killing activity[42]. Together, these data demonstrated that *Cdk4* and *Cdk6* deficiency in cancer cells triggered type I IFNs responses.

Mechanistically, our results showed that *Cdk4* and *Cdk6* deficiency induced DNA damage, which activated STING-dependent Type I IFN response and anti-tumor immunity. Our analysis by GSEA revealed that "DNA sensing pathway" gene sets as "hallmark" signatures were upregulated in *Cdk4$^{-/-}$* as compared with vector control MCA205 cancer cells (Fig. 4h). Phosphorylation of H2AX at Ser 139 (γ-H2AX) is the most sensitive marker that can be used to examine damaged DNA[32]. As expected, the levels of γH2AX and cytoplasmic dsDNA were found increased in *Cdk4$^{-/-}$* and *Cdk6$^{-/-}$* cancer cells (Figs. 4 and 6). Analysis of RNA-seq data of *Cdk4$^{-/-}$* and *Cdk6$^{-/-}$* cancer cells revealed that the MCM and DNA polymerases are downregulated in *Cdk4$^{-/-}$* cells while DNA polymerases were decreased in *Cdk6$^{-/-}$* cells (Fig. 7d), which was validated by qPCR assay (Fig. 7e). There is another possibility, though we did not confirm it in our experimental setting, that DNA damage response may be compromised while CDK4/6 inhibition as shown in another study[43].

Moreover, inactivation of STING in *Cdk4$^{-/-}$* and *Cdk6$^{-/-}$* cancer cells reversed the hyper-activation of type I IFN response

and the anti-tumor effect of *Cdk4/6* knockout. The role of anti-tumor immunity triggered by CDK4/6 inhibitor has been previously reported by Goel et al.[37] through activation of type III IFN response by increased intracellular levels of endogenous retroviral double-stranded RNA. However, our studies showed that only inactivation of STING but not MAVS, could reverse the higher levels of ISG expression in *Cdk4$^{-/-}$* and *Cdk6$^{-/-}$* cancer cells (Figs. 4 and 6). In a separate study, we found that *Cdk2$^{-/-}$* cancer cells had increased intracellular levels of endogenous retroviral double-stranded RNA and hyper-activation of interferon response through the MAVS-dependent signaling pathway[25]. Interestingly, recent studies showed that abemaciclib used by Goel et al. not only inhibited CDK4/6 but also inhibited CDK2. Although we have not studied the role of type III IFN in *Cdk4$^{-/-}$* and *Cdk6$^{-/-}$* tumor growth, we have transplanted *Cdk4$^{-/-}$* and *Cdk6$^{-/-}$* cancer cells into *Ifnar1$^{-/-}$* mice. We found that *Cdk4$^{-/-}$* tumor could grow at a similar rate as vector cells and *Cdk6$^{-/-}$* tumor could grow to bigger size in *Ifnar1$^{-/-}$* mice than those in WT C57 mice, suggesting the important role of IFNAR-1 in TME for the anti-tumor immunity activated in *Cdk4$^{-/-}$* and *Cdk6$^{-/-}$* cancer cells (Figs. 3h and 5j).

Based on these studies, we have proposed a working model suggesting that *Cdk4/6* deficiency or CDK4/6 inhibition in cancer cells triggers DNA damage and subsequent activation of the cGAS-STING-dependent type I IFN production, which activates IFNAR1 and induce ISG expression to enhance immune cell infiltration and anti-tumor immune responses in TME. Future research should focus on screening sensitive cancer types and exploring biomarkers that predict CDK4/6 inhibitor efficacy in more large-scale prospective clinical trials as well as exploring the combination therapy of CDK4/6 inhibitors and immune checkpoint inhibitors for the precise treatment of malignancies.

## Materials and methods

**Patients' tumor tissues and clinical data.** This study recruited 125 breast cancer patients for immunohistochemical (IHC) analysis and 10 breast cancer patients for RNA-seq analysis from the Affiliated Tumor Hospital of Nantong University. Tumor tissues collected from surgical patients were fixed with formalin and embedded with paraffin and then examined. Patients' PBMC were collected from patients taking or not taking CDK4/6 inhibitors and then RNA was extracted for RNA-seq assay. The study was conducted according to the principles of the Declaration of Helsinki and approved by the Human Research Ethics Committee of the Affiliated Tumor Hospital of Nantong University with the ethic number of 2022-039 and we have obtained informed consent from all participants. All ethical regulations relevant to human research participants were followed.

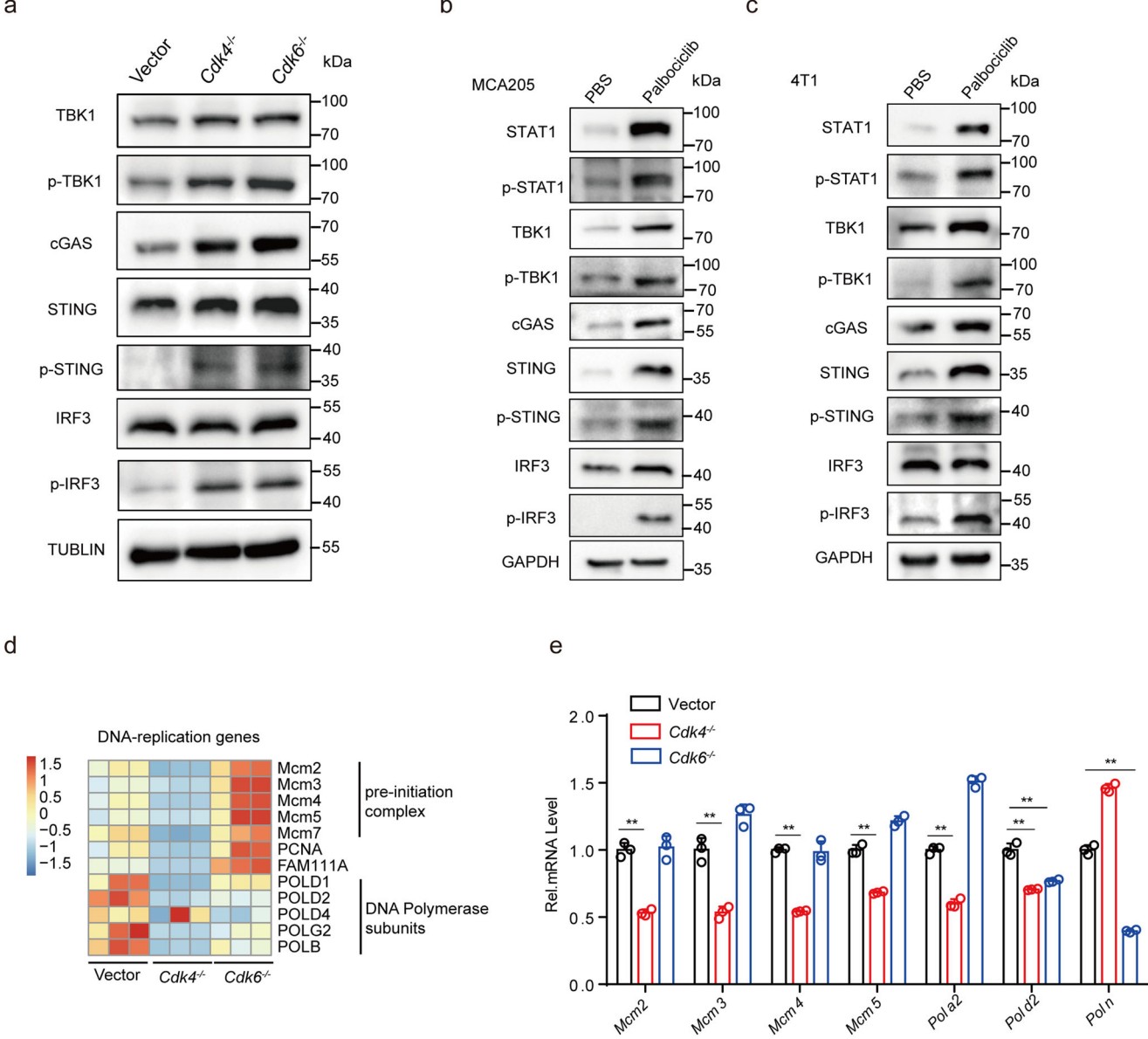

**Fig. 7 cGAS-STING pathway activated by DNA damage in *Cdk4*$^{-/-}$ and *Cdk6*$^{-/-}$ cells. a** Protein expression of TBK1, p-TBK1, cGAS, STING, p-STING, IRF3 and p-IRF3 in vector, *Cdk4*$^{-/-}$ and *Cdk6*$^{-/-}$ MCA205 cells determined by Western blot assay. **b** Protein expression of STAT1, p-STAT1, TBK1, p-TBK1, cGAS, STING, p-STING, IRF3 and p-IRF3 in MCA205 cells treated with PBS or palbociclib (0.5 μM) for 10days. **c** Protein expression of STAT1, p-STAT1, TBK1, p-TBK1, cGAS, STING, p-STING, IRF3 and p-IRF3 in 4T1 cells treated with PBS or palbociclib (0.5 μM) for 10days. **d** Heat map of mRNA expression involved in "DNA replication process" in vector, *Cdk4*$^{-/-}$ and *Cdk6*$^{-/-}$ MCA205 cells. **e** mRNA expression of *Mcm2*, *Mcm3*, *Mcm4*, *Mcm5*, *Pola2*, *Polg2* and *Poln* by RT-PCR in vector, *Cdk4*$^{-/-}$ and *Cdk6*$^{-/-}$ MCA205 cells. Data are representative of three independent experiments and presented as mean ± SD. Statistical significance was analyzed by unpaired Student's *t*-test. ns, no significant, **$p < 0.01$.

**Immunohistochemical (IHC) analysis of breast cancer patients.** Tumor tissues collected from surgical patients were fixed with formalin, embedded with paraffin and cut into 4μm pieces. The slices were transferred to 65°C incubator overnight after baking. Then they were soaked in xylene 10 min×3 times for dewaxing, in anhydrous ethanol for 3 min×3 times, in 95% ethanol for 3 min×2 times and in 75% ethanol for 3 min×3 times to remove excess liquid. Next they were rinsed with distilled water for 1 min and put in PBS. The slices were placed in citrate buffer for antigen repair. Add 3% $H_2O_2$ and incubate at room temperature for 10 min to block endogenous peroxidase. Add 100 μL primary antibody (Anti-CDK4, Anti-CDK6), incubate at room temperature for 1 h and wash with PBS buffer for 3 min×3 times. Secondary antibody was incubated at room temperature for 30 min.

Add freshly prepared DAB chromogenic solution and incubate at room temperature for 5 ~ 8 min. Rinse with tap water and recolor with hematoxylin dye solution. Then dehydrate with ethanol, transfer to xylene for 15 min and seal with neutral gum.

**Cell culture.** TC1 (murine lung epithelial cell line) and HEK293T (human embryonic kidney cell line) cell lines were obtained from Cell Resource Center, Institute of Basic Medical Sciences, Chinese Academy of Medical Sciences (Beijing, China). MCA205 (murine fibrosarcoma cell line) was obtained from Dr. S. A. Rosenberg (NCI, Bethesda, MD). All cell lines were cultured in Dulbecco's modified Eagle's medium (DMEM) (Gibcol, C11995500BT) supplemented with 10% FBS (Excell Bio, FSP500) and 100 U/mL penicillin and streptomycin (Gibcol,15140163). All cell lines were

**Table 1 Expression of CDK4 and CDK6 of different clinical pathological characteristics for breast cancer patients.**

| Parameters | Breast Cancer patients ($n=125$) | CDK4 IHC score (Mean ± SD) | CDK6 IHC score (Mean ± SD) |
|---|---|---|---|
| Vascular tumor thrombus, n (%) | | | |
| Present | 41 (32.8) | 8.78 ± 2.46 | 9.20 ± 2.38 |
| Absent | 81 (64.8) | 9.01 ± 2.35 | 9.46 ± 2.07 |
| missing | 3 (2.4) | 8.67 ± 3.51 | 10.00 ± 1.73 |
| Lymph node metastasis, n (%) | | | |
| Present | 69 (55.2) | 9.19 ± 2.32 | 9.34 ± 2.16 |
| Absent | 54 (43.2) | 8.65 ± 2.34 | 9.41 ± 2.20 |
| Missing | 2 (1.6) | 7.50 ± 6.36 | 10.50 ± 0.71 |
| Tumor size, n (%) | | | |
| ≥3 cm | 31 (24.8) | 9.10 ± 2.48 | 9.17 ± 2.10 |
| <3 cm | 93 (74.4) | 8.85 ± 2.38 | 9.47 ± 2.19 |
| Missing | 1 (0.8) | 11 | 8 |
| Tumor differentiation, n (%) | | | |
| High differentiation | 13 (10.4) | 8.08 ± 2.40 | 8.69 ± 1.89 |
| Middle differentiation | 55 (44) | 9.38 ± 2.29 | 9.80 ± 2.07 |
| Low differentiation | 46 (36.8) | 8.61 ± 2.34 | 8.98 ± 2.42 |
| Missing | 11 (8.8) | 9.00 ± 2.93 | 9.91 ± 1.22 |
| Prognosis, n (%) | | | |
| Survive | 112 (89.6) | 8.87 ± 2.42 | 9.41 ± 2.13 |
| Death | 4 (3.2) | 8.75 ± 0.96 | 7.75 ± 2.50 |
| Missing | 9 (7.2) | 9.78 ± 2.49 | 9.89 ± 2.32 |
| Receptor expression, n (%) | | | |
| Triple negative | 22 (17.6) | 8.68 ± 2.57 | 9.23 ± 2.12 |
| Triple positive | 10 (8) | 9.30 ± 2.16 | 8.56 ± 2.30 |
| HR + /HER2- | 61 (48.8) | 8.85 ± 2.48 | 9.41 ± 2.28 |
| HR-/HER2+ | 20 (16) | 8.90 ± 2.27 | 9.95 ± 2.01 |
| Others | 5 (4) | 8.20 ± 2.17 | 9.80 ± 1.30 |
| Missing | 7 (5.6) | 10.00 ± 2.16 | 9.43 ± 2.51 |

cultured in a humidified chamber at 37 °C, with 5% $CO_2$. All culture products were purchased from Gibcol. Mycoplasma were tested using cell culture supernatant by PCR assay. Normocin (InvivoGen, catalog no. ant-nr-1) and plasmocure (InvivoGen, catalog no. ant-pc) were used for the prevention and treatment of mycoplasma contamination respectively. All experiments were conducted without mycoplasma contamination.

**Tumor models**. 6-8-week female mice were used for all animal experiments. C57BL/6 and athymic nude BALB/c mice (nu/nu) were purchased from Beijing Vital River Company. Ifnar1$^{-/-}$ C57BL/6 mice were purchased from Model Animal Research Center of Nanjing University. NOD.Cg-Prkdcscid IL2rgtm1Wjl/ SzJ mice (NSG) were purchased from Beijing Biocytogen Company. Animal experimental protocols were approved by the Institutional Animal Care and Use Committee (IACUC) of Suzhou Institute of System Medicine. We have complied with all relevant ethical regulations for animal testing. All purchased mice were acclimated circumstance for one week prior to all experiments. In mice xenograft model, $2 \times 10^6$ of different tumor cells suspended in 100 μl PBS per mouse were transplanted subcutaneously. Chemotherapy was administered when tumor size reaches 25 to 45 mm$^2$ (normally 5 to 7 days after tumor cell transplantation). 5 mM palbociclib (Selleck, S1579) or PBS was administered 50 μl per mouse by intratumoral injection for three consecutive days. In CD8$^+$ T cells depletion experiment, 200 ug anti-CD8 antibody (BioXCell, BE0004-1) was injected intravenously at the indicated time points. Tumor size was monitored

2-3 times per week using calipers. Tumor size was displayed with multiplication of tumor length and width and data were described as error bars of mean ± SD. Mice were euthanized by $CO_2$ inhalation when tumor size reached to 300 mm$^2$. Tumors were harvested on day 9-11 post-implanted for following analysis such as RNA sequencing, ELISPOT assay, immunofluorescence staining and flow cytometry analysis.

**Construction of knock-out cell lines with CRISPR/Cas9 system**. SgRNA sequences targeting *Cdk4*, *Cdk6*, *Sting*, *Mavs* (primer sequences are listed in Supplementary Table 1) were designed and synthesized, then cloned into LentiCRISPR V2 vector plasmid (Addgene, #52961). Lentivirus was obtained from HEK293T cells by co-transfection of psPAX2 plasmid (Addgene, #12260), pMD2.G plasmid (Addgene, #12259) and *Cdk4*-sgRNA plasmid, *Cdk6*-sgRNA, Mavs-sgRNA plasmid, *Sting*-sgRNA plasmid or control vector plasmid with Lipofectamine 3000 (Thermo Fisher Scientific) according to the provider's protocols for 48 hours. After passing through a 0.22 μm filter, lentivirus supernatant was collected and then stored at −80 °C. The lentivirus stocks were used to infect TC-1 cells or MCA205 cells. After 48 h infection, cells were cultured in selective medium containing 4ug/ml puromycin (InvivoGen, #ant-pr-1) for at least one week. Monoclonal cells were obtained from FACSAria™ III cell sorter (Becton Dickinson, San José, CA, USA) and cultured in 96-well plate. The knock-out cell lines were identified by Western Blot assay.

**Flow cytometry analysis of tumor-infiltrating immune cells**. Mice were euthanized by $CO_2$ inhalation and tumor tissues were took out and cut into small pieces with surgical scissors in serum-free RPMI 1640 media. Then tumor tissues were digested with DNase I (Sigma, 260913-10 MU) and Liberase TL (2 μg/mL, Roche, 05401020001) at 37 °C incubator for at least 1 h. The cell suspension went through a 70 μm filter (ThermoFisher Scientific) and pelleted by centrifuge at 1500 rpm for 5 min. Red cells were removed with red cell lysis buffer. Then pelleted cells were washed and resuspend in PBS containing dye for flow cytometric analysis.

Live cells were strained by fluorescence-labeled antibodies against vivid yellow (Invitrogen, #L34959). Cell surface markers were stained by mouse specific antibodies CD45.2 (1:100, BioLegend, 109828), CD8 (1:100, BioLegend, 100723), CD11b (1:100, BioLegend, 101228), CD11c (1:100, BioLegend, 117318), IA/IE (1:100, BioLegend, 107616) at 4 °C for 30 min. For GZMB and IFN-γ staining, cells were fixed with infixation/permeabilization kit (BD Bioscience, 554714) at 4 °C for 30 min, then stained with anti-granzyme B (1:100, BioLegend, 515406) and anti-IFN-γ (1:100, BioLegend, 505813). All these antibodies were diluted by 1:100 using PBS. Stained cells were washed by PBS once and then conducted using Attune NxT Flow Cytometer (Thermo Fisher Scientific) and BD LSRFortessa, analyzed with flowjo software (Tree Star, Inc., Ashland, OR, USA).

**Flow cytometry analysis of cell cycle**. Cells were collected after inoculated for 48 h in 10 cm$^2$ cell culture dishes. Cells were washed twice with PBS and resuspended by adding 300 μl of PBS, and then mixed slowly with 700 μl of anhydrous ethanol. PI (BD Bioscience, 550825) staining was performed on ice for 30 min (avoid light) after cells were fixed at 4 °C for 24 h. Cells were washed by PBS and conducted with BD LSRFortessa, analyzed with flowjo software (Tree Star, Inc., Ashland, OR, USA).

**Protein extraction and immunoblotting**. Whole cell extracts were prepared by cell lysis buffer (Beyotime, #P0013) adding Protease and Phosphatase Inhibitor Cocktail (1:100, NCM).

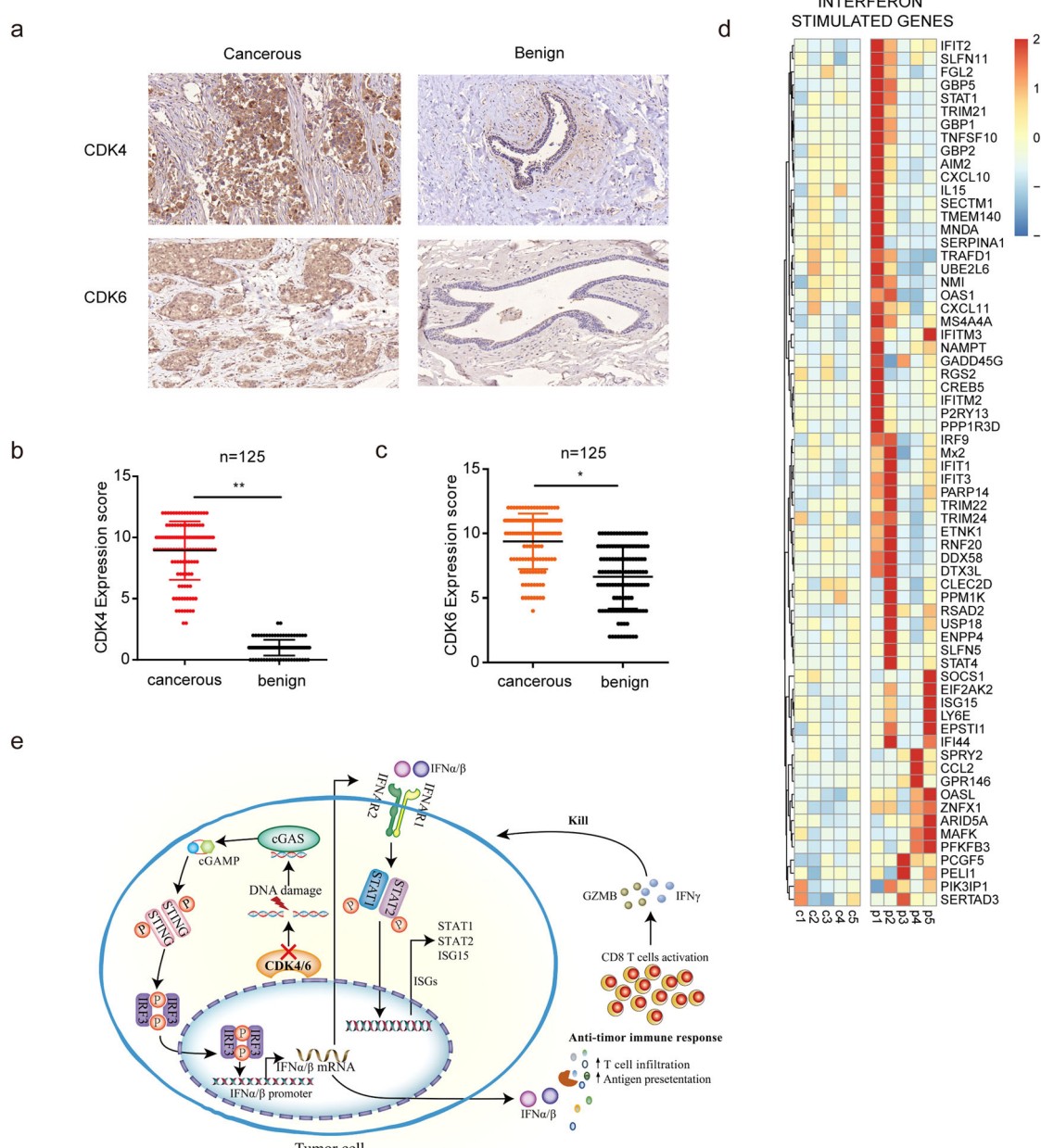

**Fig. 8 CDK4 and CDK6 is highly expressed in breast cancers and ISGs were upregulated in patients treated with palbociclib. a** Expression of CDK4 and CDK6 determined by immunohistochemical in cancerous and benign tissue of breast cancers. **b** The expression of CDK4 (**b**) and CDK6 (**c**) detected by tissue microarray immunohistochemical in cancerous and benign breast cancer tissues (*n* = 125). **d** Heat map of mRNA expression of "Interferon stimulated genes" in PBMC of breast cancer patients (*n* = 5) with or without palbociclib treatment. **e** Proposed CDK4 and CDK6 blockade induced anti-tumor pathway schematic. Data are presented as the mean ± SD. \*\**p* < 0.01. Statistical significance was determined by the Mann–Whitney *U* test.

**Table 2 Information of breast cancer patients for RNA-seq analysis.**

| Patient number | Age | TNM stage | HER2 expression | ER expression | PR expression | Tumor differentiation |
|---|---|---|---|---|---|---|
| C1 | 55 | IV | ++ | +++ | ++ | II-III |
| C2 | 50 | IV | − | +++ | +++ | II |
| C3 | 54 | IV | − | ++ | +++ | I |
| C4 | 55 | III | − | +++ | +++ | II |
| C5 | 64 | IV | − | ++ | + | II |
| P1 | 62 | IV | − | +−++ | +++ | II |
| P2 | 54 | IV | +− | + | − | II |
| P3 | 68 | IV | − | + | + | II-III |
| P4 | 39 | IV | − | ++ | + | II |
| P5 | 38 | IV | − | + | + | II |

Protein concentrations were determined using BCA Protein Quantitative kit (NCM). Equal amount of protein samples were electrophoresed using 10–12.5% SDS-PAGE gels (Epizyme Biotechnology, 02471200) and then transferred onto a 0.22 μm or 0.45 μm PVDF membrane (Bio-Rad). Blots were blocked in 5% non-fat milk (Solarbio) at room temperature for 1 h. Blots incubated with specific primary antibodies: Anti-CDK4 (1:1000, abcam, ab199728), Anti-CDK6 (1:1000, abcam, ab241554), Anti-STAT1 (1:1000, CST, 14994), Anti-Phospho-STAT1(1:1000, CST, 9167), Anti-STAT2 (1:1000, CST, 72604), Anti-Phospho-STAT2 (1:1000, CST, 88410 S), Anti-MAVS (1:1000, CST, 4983), Anti-STING (1:1000, CST, 13647), Anti-Phospho-STING (1:1000, CST, 72971), Anti-TBK1 (1:1000, CST, 38066), Anti-Phospho-TBK1 (1:1000, CST, 5483), Anti-cGAS (1:1000, CST, 31659), Anti-IRF3 (1:1000, CST, 4302), Anti-Phospho-IRF3 (1:1000, CST, 29047), Anti-β-ACTIN (1:4000, CST, 4970), Anti-GAPDH (1:4000, CST, 5174), Anti-Phospho-H2AX (1:1000, CST, 9718), Anti-Phospho-CHK1 (1:1000, CST, 12302), Anti-β-TUBLIN (1:1000, CST, 2148) at 4 °C overnight. The next day, blots were washed by 1×TBST for three times and incubated with HRP-linked secondly Antibody (1:10000, CST, 7074) at room temperature for 1 h. The membranes were scanned with the ChemiDoc XRS+ system (Bio-Rad, USA) and analyzed with Image lab software.

**RNA extraction and RT-qPCR**. Cell total RNA was extracted using the RNA-Quick Purification Kit (ES Science) according to the protocol. Extracted RNA (≤5 μg) was transcribed into cDNA using Two Step PrimeScript RT-PCR kit (TaKaRa, 6210 A). The cDNA samples were diluted with nuclease-free water into suitable concentration and used for real-time quantitative PCR (RT-qPCR) assay. cDNA amplification were conducted by SYBR Green qPCR mix (Bimake, B21202) and detected using the LightCycler480II Real-Time PCR System (Roche). Gene-specific primer sequences are listed in Supplementary Table 2. The data were normalized to ACTIN and presented as fold change of mRNA expression in experimental samples compared to the control sample.

**RNA sequencing**. Cells and grinded tumor tissues total RNA was extracted using RNA-Quick Purification Kit (ES Science) according to the manufacture's instruction. 500 ng RNA was used to prepare RNA-seq libraries using NEBNext Ultra II Direcrional RNA library Prep Kit (NEB, cat# E7760L). Equal quantities of cDNA were mixed for next sequencing (GENEWIZ, Suzhou, China). Raw data are available in NCBI database with the access number PRJNA893858 and in CNCB with the access number HRA003815.

**Gene Ontology and Gene set enrichment analysis (GSEA)**. Gene Ontology analysis was performed with David website (https://david.ncifcrf.gov/) and GSEA analysis was performed with GSEA 4.1.0 software. The standardized RNA sequencing counts were treated as input file. There were two groups for analysis of expression count matrix: WT-Low group and *Cdk4*-KO-High group or WT-Low group and *Cdk6*-KO-High group. All genes were sorted according to the degree of differential expression.

**Immunofluorescence staining**. Tumor tissues were harvested 7–10 days after subcutaneous transplantation and fixed with 4% polyformaldehyde (Biosharp, BL539A) for 24 h at 4 °C, dehydrated in 30% (wt/vol) sucrose solution for 48 hours and then embedded in OCT at room temperature. Tissue sections were cut to a thickness of 5μm with Leica microtome (Leica, CM1950). For cell samples, $3 \times 10^4$ cells were grown immunofluorescence chamber overnight. The next day the chambers were washed with PBS and then fixed in 4% paraformaldehyde at room temperature for 15 min. Samples were then permeabilized with 0.1% Triton X-100 for 5 min and treated with 10% FBS for 1 hour at room temperature to block nonspecific sites of antibody. Samples were incubated with primary antibodies: Anti-cleaved-caspase-3 (1:100, CST, 9664), Anti-CD8a (1:100, Abcam, ab217344), Anti-γH2AX (1:100, CST, 7631), Anti-cGAS (1:100, Thermo Fisher Scientific, PA5-141097), Anti-dsDNA (1:250, Santa Cruz, sc-58749) at 4 °C overnight. The next day, the chambers were washed with PBS for three times and incubated with corresponding secondary antibody (Biolegend) at room temperature for 2 h. Then chambers were incubated with Hochest (Thermo Fisher Scientific, H3569) at room temperature for at least 15 minutes and then washed with PBS. Anti-quenching agent was added to the chambers to avoid fluorescence quenching. All staining process was avoid of light. Images were captured with confocal microscope (Leica TCS SP8) and analyzed by ImageJ software.

**Enzyme-linked immune spot (ELISpot) assay**. IFN-γ production in tumor microenvironment was detected with IFN-γ ELISPOT assay kit (BD Biosciences, 551881). Tumor tissues were harvested at 8-12 days after transplantation. Tumor tissues were digested with DNase I (Calbiochem) and Liberase TL (Roche) at 37 °C for at least 1 h. The cell suspension was passed through a 70 μm filter (ThermoFisher Scientific) and pelleted by centrifugation at 1500 rpm for 5 min. The cell pellets were resuspended by red blood cell lysate (Solarbio) on ice for 15 minutes to remove red blood cells. Then cell suspension was centrifuged at 1500 rpm for 5 min and washed by PBS twice. $2 \times 10^6$ cells were seeded per well in a pre-coated with purified anti-mouse IFNγ antibody (1:200) plate (EMD Millipore) and incubated incubator with 5% $CO_2$ at 37 °C for 16-24 hours. The next day, cell supernatant was removed, and the plate was washed by 1640 medium containing 10% FBS once and then blocked with 1640 medium to block nonspecific sites of antibody. Then plate was treated with biotinylated anti-mouse IFNγ antibody (BD, #511818KZ) for 2 h and streptavidin-HRP for 1 h respectively. The reaction stopped by adding final substrate solution. The plate was scanned using CTL ImmunoSpot® S6 Analyzers (CTL and CTL Analyzers, LLC) and red dots were counted after washed with ddH$_2$O and completely dry.

**Enzyme-linked immunosorbent assay (ELISA)**. IFN-β released in cell supernatant was detected with Verikine Mouse IFN-β ELISA Kit (PBL, #42400-1) according to the manufacture's protocol. Data were obtained from SpectraMax device (Molecular Devices).

**Type I IFN expression luciferase assay**. Type I IFN was detected using ISRE-L929 reporter cell line (a gift from Jiang Zhenfan laboratory, Peking University). $2.5 \times 10^4$ ISRE-L929 cells were plated per well in a 96-well cell plate overnight. The next day cell culture medium was replaced with vector or *Cdk4*−/− MCA205 cell supernatant. Cells were incubated in incubator with 5% $CO_2$ at 37 °C for additional 4 h and luciferase activity was detected in cell lysates using a luciferase assay kit (Progema).

**Cell proliferation assay**. Cells were plated with 1500 per well in 96-well cell culture plate for 24, 48, 72 and 96 hours. Cell proliferation was detected using Cell Counting Kit-8 (Dojingdo, code CK04) according to the manufacture's recommendations.

**Colony formation assay**. Cells were plated with 1000 per well in 6-well cell culture plates by flow sorting for 10 days. Then cells were fixed with 4% PFA for 15 min and stained with 0.5% crystal violet solution for 30 min and washed by PBS for three times. Cell colonies were counted when the plates were totally dry.

**Statistics and reproducibility**. Statistical analyses were performed using GraphPad Prism 7 software. Results were presented as mean ± SEM or mean ± SD of at least 3 parallel experiments. Unpaired Student's t-test or one-way ANOVA test were used as indicated for comparisons between two groups. For in vivo studies, groups were compared by Mann–Whitney $U$ test. $*p < 0.05$, $**p < 0.01$, ns (non-significant).

**Reporting summary**. Further information on research design is available in the Nature Portfolio Reporting Summary linked to this article.

## Data availability

The RNA-seq raw data in this publication are available in NCBI database with the access number PRJNA893858 and in CNCB with the access number HRA003815. Supplementary Data file includes the source data corresponding to the graphs presented in the main figures and supplemental figures. Uncropped and unedited Western blots are provided in Supplementary Fig. 4. All relevant data are available from the corresponding authors upon reasonable request.

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

## Acknowledgements

This project is financially supported by Chinese Academy of Medical Sciences Innovation Fund for Medical Sciences (2021-1-I2M-047, 2022-I2M-2-004), National Natural Science foundation of China (82073181, 81802870, 82102371 and 81874067), the Non-profit Central Research Institute Fund of Chinese Academy of Medical Sciences (2020-PT310-006), the Key Project of Jiangsu Provincial Health Commission (K2019021), the Suzhou Municipal Key Laboratory (SZS2023005), Natural Science foundation of Jiangsu Province (BK20211554).

## Author contributions

H.F., H.Y., and G.C. designed the study. H.F., W.L., Y.Z., Y.Z., J.M., H.L., Y.S. and R.C. performed the experiments including in vivo studies, western blot assay, qPCR assay and flow cytometry. H.F., L.Y. and M.G., L.L. and J.R. performed the data analysis. J.C. and X.J. provided human samples and conducted tumor tissue IHC assay. H.F., W.L., H.Y.,

and G.C. wrote and revised the manuscript. All authors read and approved the final manuscript.

## Competing interests

The authors declare no competing interests.
