## [Peer Review File · Communications Biology]

Reviewers' comments:

Reviewer #1 (Remarks to the Author):

In the manuscript by Fan et al. the authors show CDK4 and CDK6 are highly expressed in many cancers, including breast cancer. The expression of CDK4 and CDK6 is associated with poor patient outcome and reduced T cell infiltration. The manuscript shows that similarly to CDK4/6 inhibition, ablation of CDK4 or CDK6 caused tumour regression in mice which is facilitated by an immune response. The immune response is triggered when CDK4 and CDK6 is knocked out due to elevated DNA damage. This increased DNA damage activates the cGAS-STING pathway that subsequently triggers a type I interferon immune response. Additionally, they show that the knockout of CDK6 can inhibit tumour growth without a reliance on an immune response. This work is very relevant and has important implications to understand how CDK4/6 inhibition can elicit an immune response. However, there are a few issues that need to be addressed to strengthen the manuscript.

Major points:

1. The authors state that due to off targets effects of CDK4/6 inhibitors anti-tumour immunity cannot be studied accurately. The manuscript then goes on to highlight that due to their findings, CDK4/6 inhibitors may enhance anti-tumour immunity. Whilst the knockout of CDK4 and CDK6 is useful to unpick differences mechanistically here it is not as clinically relevant. Palbociclib should be used to show an increase in DNA damage and subsequent cGAS-STING pathway activation in this model and this would strengthen the point that CDK4/6 inhibitors can modulate anti-tumour immunity.
2. The data presented in Fig 6H and I, suggest that the DNA damage is a result of loss of MCM and DNA polymerases. It has also recently been shown that CDK4/6 inhibition results in the downregulation of MCM proteins contributing to replication stress (Crozier et al., 2022). However, this result is only observational in this model and should be validated. Are MCM/DNA polymerases down regulated to enough of an extent which would cause issues with DNA replication/origin licensing? The source of DNA damage should be elucidated as this is an important point which requires more mechanistic detail.
3. The data in Figure 5 shows that knockout of CDK6 behaves differently. Unlike CDK4, the ablation of CDK6 can reduce tumour burden in immunocompetent mice. Is this difference due to fact that CDK6^{-/-} cells to proliferate more slowly and perhaps these cells are becoming senescent as they are more reliant on CDK6 activity for proliferation? Or perhaps the CDK4 or CDK6 knockout have different levels of DNA damage (a side-by-side comparison of these quantifications should be included). It is also crucial to test if this difference between CDK4 and CDK6 is cell line specific to MCA205. The authors should try to recapitulate this work in TC1 cell line (or another relevant model).
4. CDK4/6 inhibitors modulate many different anti-tumour responses. It has been shown that CDK4/6 inhibitors can induce senescence which promotes a senescence associated secretory phenotype

(Wang et al., 2022). Could the immune responses seen here be partially facilitated by a SASP response following the knockdown of CDK6 or CDK4 (or with palbociclib)? The authors should address if a SASP response is seen in this model.

5. The authors state that cGAS-STING pathway can recognise cytosolic DNA to stimulate type I interferon pathway. From this data, it is clear that STING is playing a role in this response, but the authors do not show that the cytosolic DNA is recognised by cGAS. Western blots probing for cGAS should be performed or alternatively, immunofluorescence showing co-localisation of cGAS and Tunnel positive cells. cGAS may not be the sensor in this model as there may be non-canonical activation of the STING pathway (Unterholzner & Dunphy, 2018).

Minor points:

6. A very high dose of palbociclib is used in Figure 1A and B. Do the authors anticipate that there would be any off-target effects that may be contributing to these results?

7. In Figure 4J a Tunnel assay (which is a marker for fragmented DNA) is used which the authors conclude that there is an increase of Tunnel positive cells in CDK4 deficient cells. A quantification is required here to show that this fragmented DNA is cytoplasmic. Is there an increase in micronuclei with CDK4 and CDK6 deficiency that would lead to cGAS-STING pathway activation?

8. In Figure 3D an F cells are treated with palbociclib for 10 days and this increases the STAT1 and pSTAT1 and ISGs. After treatment with palbociclib are they cells arrested in G1? Or are they unable to hold a penetrant G1 arrest? If the cells are fully arrested in G1 at the 10 day timepoint, the increase in type I interferon pathway may not be due to DNA damage (as G1 arrested cells should be protected from DNA damage until they progress through the cell cycle).

9. In Figure 6H and I, this shows that MCM and DNA polymerases are downregulated which is contributing to the DNA damage phenotype. Another possibility is that the DNA damage response may be compromised as also shown in another study (Salvador-Barbero al., 2020).

Reviewer #2 (Remarks to the Author):

This manuscript investigates the individual contributions of loss of CDK4 and CDK6 to the immune effects reported to be triggered by CDK4/6 inhibitors such as Palbociclib. The approach used here was to delete CDK4 and CDK6 individually in mouse tumour cells. The data produced is somewhat

unexpected and interesting, but I think the actual models are flawed. They provide insights in the individual effects of removing CDK4 and CDK6, but as the CDK4/6 inhibitors inhibit both targets equally and simultaneously, this is a very different outcome from individual deletion. Another issue is that the RB status of the cell lines used here is not provided. The primary determinant of sensitivity to CDK4/6 inhibitors is RB status, RB wild type cells are very sensitive ($IC_{50} < 0.5 \mu M$) to these drugs, loss of RB results in >10 fold increase in IC_{50} . The authors do show that CDK4 deletion has very modest effects on in vitro proliferation, CDK6 deletion has a stronger effect, although neither reduce proliferation by 50%. RB wild type cells block in G1 phase and will eventually senesce with CDK4/6 inhibition. Without more detailed analysis of the effects of the individual deletion on cell cycle progression in these cell models, it is difficult to understand a lot of the data presented.

The authors do show that CDK4 deletion promotes a STING dependent type I interferon response that recruits and activates CD8+ T cells, and this is major component of the anti-tumour response to this tumour. Interestingly, CDK6 deletion also reduced in vivo tumour growth significantly, and although it was also shown to produce a STING-dependent interferon response, this was not dependent on B, T or NK cells, but was high dependent on the interferon response of the recipient mouse. No discussion of what recipient mouse cells were involved in this anti-tumour effect was provided. I found this part difficult to understand and the authors offered little guidance on what stromal cells were responsible the interferon-dependent effect.

A major point made in this manuscript was that CDK4 or CDK6 deletion or inhibition was promoting DNA damage. The authors did show limited evidence of increased $\gamma H2AX$ staining, although the discreet DNA damage foci being alluded to in the text were not apparent in the images shown. They went on to show that in the deleted cells there was down regulation of components of the DNA replication fork complex that could increase replication stress, a process that also increases $\gamma H2AX$ levels in cells although the staining pattern is a more general nuclear stain as opposed to discreet foci. It was the former pattern that was apparent in the images presented. It suggests that deletion of each individual CDK was promoting replication stress, albeit by slightly different routes, and it is this increased replication stress that is responsible for increase DNA damage markers and increased STING pathway function. Directly demonstrating that STING is activated would add to the manuscript.

Overall, the data present was interesting, but I do not see that it directly translates to an improved understanding of the immunostimulatory effects of CDK4/6 inhibitors. CDK4/6 inhibitor treated RB wild type cells are arrested before entering S phase and therefore cannot by definition increase replication stress. I am unsure what the effect of CDK4/6 inhibitors in RB defective tumours is, although CDK4 has other cell cycle roles outside of RB regulation.

Specific points:

What is the RB status of the mouse tumour cell lines used here? Without the RB context it is difficult to understand what the effects of deletion of CDK4 and CDK6 might be. The relative lack of effect of the individual CDK4 and CDK6 suggest that these cell lines could be RB mutant/defective. Does Palbociclib inhibit these mouse lines in vitro and what is the effect of inhibition, G1 arrest and senescence or something else?

Figure 11; This is an interesting finding, but many controls are missing. What was tumour growth like in the initial vaccination sites? Did these tumours grow normally, or at least as expected? If a similar experiment was performed using the WT cells, would a similar rejection be observed?

The tumour growth data is presented as mean +/-SEM. This does not provide a good indication of the scatter of the data, especially with small sample sizes (n=5). SD would be more appropriate.

Figure 2C; The gating strategy is not clear the Supplementary figure 2 but the CD8+ population in the CDK4 deleted tumours appears to be two populations, a CD45hi and CD45lo population. Only the CD45lo population appear to increase in the CDK4 deleted tumours. Are these really CD8+ T cells?

Figure 3F; The very different effects of the two doses of Palbociclib are surprising. In the RB wt cells palbociclib has an IC50 of > 0.5 uM, so the big difference between two very similar doses is difficult to understand.

Figure 4J; TUNEL is also used to identify cells with cleaved DNA that is associated with apoptosis. As you have already demonstrated increased cleaved caspase 3, this TUNEL staining is more likely to be associated with apoptosis than simple DNA damage.

Figure 4K; The relatively intense gH2AX staining appears to be present in all CDK4 deleted cells. Is this a common feature? The immunoblot does not show the same difference in gH2AX levels. The staining also appears to be an even staining of the nucleus rather than discreet DNA damage foci. This more even staining has been associated with replication stress rather than discreet double strand breaks. Is this the case or are discreet foci the most common staining pattern? Better images would be necessary to see these. The STING-dependent innate immune response is triggered by cytoplasmic DNA, but this has not been demonstrated to increase in this manuscript.

Supplementary Figure 1 E, F, 3C; The data don't show a correlation. There a very small number of outliers that may be correlated, but the majority of CDk4 and CDK6 over-expressing tumours have very low infiltrations scores.

CCK8; this must be defined

Figure 5J; the data is very strong, but there is little discussion of this effect. What recipient animal stromal cells are involved in the IFN signalling response that inhibits tumour growth? It appears not be immune cells, or at least B, T or NK cells?

Figure 6E; I don't understand the STING knockout data. STING deletion does reduce IFN expression, but then why is the effect of STING deletion in CDK6 deleted cells not the same as CDK6 deleted cells in IFNAR1 deleted mice?

Figure 7E; This is a very different effect. Palbociclib will inhibit proliferation and may start to trigger senescence in PBMCs. Part of the senescence program is SASP expression which can include ISGs. As you have presented no evidence that CDK4 or CDK6 deletion promotes senescence in your models this is a quite different effect.

Reviewer #3 (Remarks to the Author):

Overall, this study by Huimin Fan et al. provides new insight into how CDK4 and CDK6 blockade, using genetically engineered Cdk4 and Cdk6 CRISPR-KOs, induces both an innate and adaptive immune response through cGAS-STING pathway activation, type I interferon gene expression, and CD8+ T-cell

infiltration and activation. The experimental methodology is robust, notably the mouse modelling is well performed. However, the following points should be addressed to complete this study:

1. It is well known that CDK4/CDK6 blockade triggers a robust anti-tumor immune response, which the authors cite. The authors suggest that they observed activation of cGAS-STING signaling because of Cdk4 KO or Cdk6 KO, and not MAVS-dependent signaling as observed by Goel et al because abemaciclib also inhibits CDK2. Does inhibition of CDK2 (for example, treatment with abemaciclib or shRNA to Cdk2) in Cdk4 KO or Cdk6 KO cells activate MAVS-dependent signaling? Does treatment with Palbociclib which has been reported to act via cGAS-STING signaling (Liu et al. *Front. Immunol* 2021) of Cdk4 KO or CDK6 KO cells have an additional effect?
2. CDK4/CDK6 blockade is currently only approved in breast cancer. The majority of the data is from MCA205 fibrosarcoma and TC1 lung. It would be important to show that CDK4/CDK6 blockade in breast tumor models activates the cGAS-STING pathway leading to immune infiltration.
3. In Figure 2I, the authors show that CD8+ T-cell depletion abrogates the phenotype of Cdk4 KO in mice. Does this also occur for Cdk6 KO?
4. Figure 2A and Figure 3A are very similar. Similarly, Figure 2B and 3B are very similar.
5. Figure 6D and E figure legends are flipped.

Dear Editor Valeria Naim,

This is a response letter for manuscript #COMMSBIO-23-1235-T. Thanks for your and the reviewers' opinions. These comments are very helpful in improving the quality of the manuscript. We have carefully considered the reviewers' suggestions, revised our manuscript, and added some results to enrich the article. Words with high brightness are modified or added content in the manuscript. The following is our point-by-point reply to the editor and reviewers' comments.

SUGGESTIONS FROM EDITOR:

1. Please provide further evidence/mechanistic data on the induction of replication stress/DNA damage and cGAS-STING pathway activation by cytoplasmic DNA.

Response: Thank you for the suggestion, we have added these results in Figures 4K, 6H, 6I, 7A and 7B, which showed that either CDK4/6 deletion or inhibition could induce cytoplasmic dsDNA (Figures 4K, 6H and 6I), and activate downstream cGAS-STING pathway in MCA205 and 4T1 cell lines (Figures 7A and B).

2. A more detailed comparison of the effect of palbociclib and CDK4/6 deletions in terms of cell cycle arrest and induction of senescence and SASP program, or different immune signaling pathways should be provided, as requested.

Response : We have added these research in Supplementary Figures 1G-I, Supplementary Figures 3G-J. The results demonstrated that both palbociclib treatment or CDK4/6 deletion could cause cell cycle arrest (Supplementary Figures 1E-G, 3F-G). We also found that unlike *Cdk4*^{-/-} cells, *Cdk6*^{-/-} cells could induce senescence and SASP program (Supplementary Figures 3H and I), which may explain why *Cdk6*^{-/-} cells proliferate more slowly than *Cdk4*^{-/-} cells.

3. Please address the criticism of reviewer #2 about the Rb status of the cell lines used.

Response: Thank you. We have added the result in Supplementary Figure 3E, which showed that RB expressed in WT, *Cdk4*^{-/-} and *Cdk6*^{-/-} cells and p-RB was slightly decreased in *Cdk4*^{-/-} and *Cdk6*^{-/-} cells compared with WT cells. Moreover, we tested the mutation status of RB1 in response to Reviewer #2.

Response to Reviewer #1

Major points:

1. The authors state that due to off targets effects of CDK4/6 inhibitors anti-tumour immunity cannot be studied accurately. The manuscript then goes on to highlight that due to their findings, CDK4/6 inhibitors may enhance anti-tumour immunity. Whilst the knockout of CDK4 and CDK6 is useful to unpick differences mechanistically here it is not as clinically relevant. Palbociclib should be used to show an increase in DNA damage and subsequent cGAS-STING pathway activation in this model and this would strengthen the point that CDK4/6 inhibitors can modulate anti-tumour immunity.

Response: We appreciate very much for your suggestion. We have added experiments according to your ideas, which are shown in Figures 6I, 7B and C. It is shown that Palbociclib could increase

the cytosolic dsDNA and subsequent cGAS-STING pathway in both MCA205 and breast cancer cell line 4T1.

2. The data presented in Fig 6H and I, suggest that the DNA damage is a result of loss of MCM and DNA polymerases. It has also recently been shown that CDK4/6 inhibition results in the downregulation of MCM proteins contributing to replication stress (Crozier et al., 2022). However, this result is only observational in this model and should be validated. Are MCM/DNA polymerases down regulated to enough of an extent which would cause issues with DNA replication/origin licensing? The source of DNA damage should be elucidated as this is an important point which requires more mechanistic detail.

Response: Thank you for the suggestion. We have added more mechanistic detail as you required, as shown in Figure 7E. As you expected, MCM families were obviously down-regulated in *Cdk4*-deleted cells, whereas DNA polymerases (such as POLG2 and POLN) were down-regulated in *Cdk6*-deleted cells.

3. The data in Figure 5 shows that knockout of CDK6 behaves differently. Unlike CDK4, the ablation of CDK6 can reduce tumour burden in immunocompetent mice. Is this difference due to fact that CDK6^{-/-} cells to proliferate more slowly and perhaps these cells are becoming senescent as they are more reliant on CDK6 activity for proliferation? Or perhaps the CDK4 or CDK6 knockout have different levels of DNA damage (a side-by-side comparison of these quantifications should be included). It is also crucial to test if this difference between CDK4 and CDK6 is cell line specific to MCA205. The authors should try to recapitulate this work in TC1 cell line (or another relevant model).

Response: These are interesting questions that are worthy of exploration. Cell proliferation assay detected by cell counting kit-8 (CCK8) assay demonstrated that *Cdk6*^{-/-} cells proliferated more slowly than *Cdk4*^{-/-} cells (Figure 5F). We further detected the senescence condition of these cells and found that unlike *Cdk4*^{-/-} cells, *Cdk6*^{-/-} cells are becoming senescent, which may explain why *Cdk6*^{-/-} cells proliferate more slowly (data were added in Supplementary Figure 3H and I). As you suggested, We have also added quantifications of γ H2AX foci percentage of vector, *Cdk4*^{-/-} and *Cdk6*^{-/-} cells (Figures 4J and 6F).

We have also generate two *Cdk6*^{-/-} TC1 cell lines to recapitulate this work and found similar results as MCA205 cell line (Figures 5D, E and H).

4. CDK4/6 inhibitors modulate many different anti-tumour responses. It has been shown that CDK4/6 inhibitors can induce senescence which promotes a senescence associated secretory phenotype (Wang et al., 2022). Could the immune responses seen here be partially facilitated by a SASP response following the knockdown of CDK6 or CDK4 (or with palbociclib)? The authors should address if a SASP response is seen in this model.

Response: Thank you for your suggestion. We have detected SASP response of WT, *Cdk4*^{-/-} and *Cdk6*^{-/-} cells. It is interesting to find that unlike *Cdk4*^{-/-} cells, *Cdk6*^{-/-} cells are becoming senescent, which may explain why *Cdk6*^{-/-} cells proliferate more slowly (as I mentioned in response to point 3). We cannot exclude the possibility that the immune responses could be partially facilitated by SASP response in *Cdk6*^{-/-} cells. However, we observed the appearance of cytosolic dsDNA, and

subsequent cGAS-STING activation in *Cdk6*^{-/-} cells (Figures 6G and 7A). Importantly, knockout of Sting could partially reverse the anti-tumor effect of *Cdk6* deficiency (Figure 6E), which exert the critical role of DNA damage-cGAS-STING pathway in *Cdk6*-deficiency induced anti-tumor immune responses. The questions you raised are very meaningful, we would like to continue to study whether SASP responses in *Cdk6*^{-/-} cells contributed to anti-tumor immune responses in our future study.

5. The authors state that cGAS-STING pathway can recognise cytosolic DNA to stimulate type I interferon pathway. From this data, it is clear that STING is playing a role in this response, but the authors do not show that the cytosolic DNA is recognised by cGAS. Western blots probing for cGAS should be performed or alternatively, immunofluorescence showing co-localisation of cGAS and Tunnell positive cells. cGAS may not be the sensor in this model as there may be non-canonical activation of the STING pathway (Unterholzner & Dunphy, 2018).

Response: Thank you for the helpful advice. We agree that the experiments you mentioned would strengthen our conclusion. The correlative results are shown in Figures 4K, 6J-I. We stained dsDNA (red) and cGAS (green) and found that they have co-localization (yellow) in *Cdk4*^{-/-} and *Cdk6*^{-/-} cells. As shown in Figure 8A, cGAS-STING pathway was activated (the expressions of cGAS, p-TBK1, p-IRF3 and p-STING were up-regulated) in both *Cdk4*^{-/-} and *Cdk6*^{-/-} cells.

Minor points:

6. A very high dose of palbociclib is used in Figure 1A and B. Do the authors anticipate that there would be any off-target effects that may be contributing to these results?

Response: Yes, as you mentioned, we cannot exclude the possibility of the off-targets, which is a common side effect of small-molecular inhibitors. However, we examined the inhibitory effect of palbociclib (0.5uM) on tumor cell growth (Supplementary Figure 1E) and its function of inducing cGAS-STING and subsequent type I interferon pathway activation (Figures 8B and C). All the results could support the conclusion that palbociclib triggers type I interferon-induced anti-tumor responses by activating cGAS-STING pathway.

7. In Figure 4J a Tunnel assay (which is a marker for fragmented DNA) is used which the authors conclude that there is an increase of Tunnel positive cells in CDK4 deficient cells. A quantification is required here to show that this fragmented DNA is cytoplasmic. Is there an increase in micronuclei with CDK4 and CDK6 deficiency that would lead to cGAS-STING pathway activation?

Response: Thank you for the useful suggestion. As pointed by reviewer 2#, Tunnel positive could also represent apoptosis. We replaced Tunnel staining results with the staining of co-localization of dsDNA and cGAS to avoid this confusion. In Figures 4K and 6G, we could clearly see the dsDNA is cytoplasmic. We did not observe there is an increase in micronuclei in *Cdk4*^{-/-} and *Cdk6*^{-/-} cells compared with WT cells.

8. In Figure 3D an F cells are treated with palbociclib for 10 days and this increases the STAT1 and pSTAT1 and ISGs. After treatment with palbociclib are they cells arrested in G1? Or are they unable to hold a penetrant G1 arrest? If the cells are fully arrested in G1 at the 10 day timepoint, the increase in type I interferon pathway may not be due to DNA damage (as G1 arrested cells should be protected from DNA damage until they progress through the cell cycle).

Response: To answer your question, we conducted flow cytometry to analyze cell cycle after treatment with 0.5uM palbociclib. As shown in the figure (which is added in Supplementary Figures 1F and G), the percentage of cells in G1 phase was increased after treatment with palbociclib for 5 days and 10 days, indicating that palbociclib could cause G1/S arrest. However, not all the cells hold a penerant G1 arrest as approximately 55% cells could still transit to S and G2/M phase.

9. In Figure 6H and I, this shows that MCM and DNA polymerases are downregulated which is contributing to the DNA damage phenotype. Another possibility is that the DNA damage response may be compromised as also shown in another study (Salvador-Barbero et al., 2020).

Response: Yes, this possibility you mentioned may exist. That is an interesting finding, we have cited this research in our discussion part (page 32, line 733-738)

Response to Reviewer #2

1. What is the RB status of the mouse tumour cell lines used here? Without the RB context it is difficult to understand what the effects of deletion of CDK4 and CDK6 might be. The relative lack of effect of the individual CDK4 and CDK6 suggest that these cell lines could be RB mutant/defective. Does Palbociclib inhibit these mouse lines in vitro and what is the effect of inhibition, G1 arrest and senescence or something else?

Response: Thank you for your constructive comments on my manuscript. We have done some correlative research to answer the questions you raised. As shown in the figures below (added in Supplementary Figure 3E), RB was expressed in WT, *Cdk4*^{-/-} and *Cdk6*^{-/-} cells and p-RB was slightly decreased in *Cdk4*^{-/-} and *Cdk6*^{-/-} cells compared with WT cells. In addition, we also detected RB mutation in these cells. There are 38 point mutations in WT cells, 6 in introns and 32 in exons, which include 25 synonymous mutations and 7 nonsynonymous mutations. In *Cdk4*^{-/-} cells, there are 32 point mutations, 5 in introns and 25 in exons, which include 20 synonymous mutations and 5 nonsynonymous mutations. In *Cdk6*^{-/-} cells, there are also 38 point mutations, 6 in introns and 32 in exons, which include 25 synonymous mutations and 7 nonsynonymous mutations. All these mutations have not been reported to be correlated with diseases.

0.5uM palbociclib could inhibit MCA205 cell proliferation as indicated by CCK8 assay (Supplementary Figure 1E). Flow cytometry analysis showed the percentage of cells in G1 phase was increased after treatment with palbociclib for 5 days and 10 days, indicating that palbociclib could cause G1/S arrest (Supplementary Figure 1F and G).

The mutation comparison is shown in Venn figure:

2. Figure 1I; This is an interesting finding, but many controls are missing. What was tumour

growth like in the initial vaccination sites? Did these tumours grow normally, or at least as expected? If a similar experiment was performed using the WT cells, would a similar rejection be observed?

Response: Tumors on initial vaccination site grew normally as they grew in WT C57 mice. We did not use WT cells as control because WT cells grow much faster than *Cdk4*^{-/-} cells. We re-challenged with WT cancer cells on the right flank two weeks after immunization and started monitoring tumor size 7 days later. So the left cells would grow 21 days when we start to monitor the tumor size on the right flank. Mice should be euthanized when tumors reach 300mm² according to animal ethics before there is a difference between groups. We still tried to add a control group as you suggested. According to other literature methods (Liu, et al, Cell Mol Immunol, 2021), we supplemented freeze-thawed WT cells, which could not form tumors at vaccination sites, as control and result is shown in Figure 1I. There were no difference between immunization with PBS and freeze-thawed WT cells groups. Immunization of mice with *Cdk4*^{-/-} tumor cells could effectively suppress WT tumor growth on the right flank.

3. The tumour growth data is presented as mean +/-SEM. This does not provide a good indication of the scatter of the data, especially with small sample sizes (n=5). SD would be more appropriate.

Response: Thank you for the helpful suggestion. We have corrected this in all tumor growth data.

4. Figure 2C; The gating strategy is not clear the Supplementary figure 2 but the CD8⁺ population in the CDK4 deleted tumours appears to be two populations, a CD45^{hi} and CD45^{lo} population. Only the CD45^{lo} population appear to increase in the CDK4 deleted tumours. Are these really CD8⁺ T cells?

Response: We feel very sorry to make the gating strategy confused. CD8⁺ T cells were sorted from total cells. We have corrected the gating strategy in Supplementary Figure 2. Thank you a lot for pointing this out.

5. Figure 3F; The very different effects of the two doses of Palbociclib are surprising. In the RB wt cells palbociclib has an IC50 of > 0.5 uM, so the big difference between two very similar doses is difficult to understand.

Response: We tried to repeat this experiment but found that the drug we used before is out of expiry (the time we used before was within expiry) so we bought another one. The newly bought

Palbociclib could activate cGAS-STING and subsequent type-I interferon pathway in both MCA205 and 4T1 cells at 0.5uM (figures are shown below). The difference of the drug effect may due to their different manufacturers. The one we used before was bought from Selleck, catalog S1116 and the one we newly bought was from MCE, catalog HY-50767. We have changed figure 3F using the result with newly bought one.

6. Figure 4J; TUNEL is also used to identify cells with cleaved DNA that is associated with apoptosis. As you have already demonstrated increased cleaved caspase 3, this TUNEL staining is more likely to be associated with apoptosis than simple DNA damage.

Response: In Figure 2, cleaved-caspase-3 was stained on tumor tissues, not cell lines. Infiltrated CD8+ T cells could kill tumor cells in tumor micro-environment, resulting in the expression of cleaved-caspase-3. Still, we agree that TUNEL positive not only identifies cells with cleaved DNA but also those undergoing apoptosis, so we replaced TUNEL staining results with the staining of co-localization of dsDNA and cGAS to avoid this confusion (Figures 4K and 6G).

7. Figure 4K; The relatively intense gH2AX staining appears to be present in all CDK4 deleted cells. Is this a common feature? The immunoblot does not show the same difference in gH2AX levels. The staining also appears to be an even staining of the nucleus rather than discrete DNA damage foci. This more even staining has been associated with replication stress rather than discrete double strand breaks. Is this the case or are discrete foci the most common staining pattern? Better images would be necessary to see these. The STING-dependent innate immune response is triggered by cytoplasmic DNA, but this has not been demonstrated to increase in this manuscript.

Response: We feel very sorry that we did not make it clear in the figure that gH2AX is also called p-H2AX. We just explained this in our manuscript (page 19, line 476-477) So the gH2AX protein level was also increased in *Cdk4*^{-/-} and *Cdk6*^{-/-} cells (Figures 4I and 6G). We have modified the figures to make it clear. Thank you for pointing this out.

We have searched some relevant literature and found that both even staining of nucleus (e.g. Marti TM et al, Proc Natl Acad Sci U S A, 2006) and discrete foci (e.g. S Burdak-Rothkamm et al, Oncogene, 2007) could be presented by gH2AX staining in different articles. And it seems that as gH2AX expression increases, the fluorescence intensity is enhanced and the fluorescence pictures transit from foci to even staining in nucleus, which consists of multiple positive foci. We have

changed our staining pictures into better quality images (Figures 4J and 6F) as you suggested. We enlarged part of the image so that the foci could be clearly observed.

8. Supplementary Figure 1 E, F, 3C; The data don't show a correlation. There are very small numbers of outliers that may be correlated, but the majority of CDK4 and CDK6 over-expressing tumours have very low infiltrations scores.

Response: We searched the correlation between CDK4, CDK6 and CD8⁺ T cell infiltration in the TIMER 2.0 website. Once the gene and immune infiltrates were submitted, a heatmap with numbers will show the purity-adjusted Spearman's rho across various cancer types. Then we click CD8⁺ T cell on the heatmap, a scatter plot will pop out to present the relationship between infiltrates estimation value and gene expression. The results are based on certain algorithms of the website. We agree that samples with low expression of CDK4/6 and high infiltration scores are too small so we decided to delete these figures.

9. CCK8; this must be defined

Response: We appreciate this helpful suggestion and the definition of CCK8 assay was added in our manuscript (page 22, line 537). Cell counting kit-8 (CCK8) assay which was applied to detect cell proliferation.

10. Figure 5J; the data is very strong, but there is little discussion of this effect. What recipient

animal stromal cells are involved in the IFN signalling response that inhibits tumour growth? It appears not be immune cells, or at least B, T or NK cells?

Response: Comparing the tumor growth curve of *Cdk6*^{-/-} cells in WT mice, nude mice and NSG mice (Figure 5A, B and C), we believe that T, B and NK cells at least played a part role in *Cdk6* deficiency induced anti-tumor immunity. So T, B and NK cells may involved in *Cdk6* deficiency mediated IFN signaling response. Though we did not evaluate the function of DCs in our study, it is reported that DCs are the most sensitive cells to type I interferon among all immune cells (Asselin-Paturel et al, J Exp Med, 2005). More importantly, systemic or local DC responses are dependent on IFN-I signaling, which greatly enhances the antigen-presenting ability of DC to activate CD8⁺ T cells (Diamond et al, J Exp Med, 2011; Fuertes et al, J Exp Med, 2011). Moreover, DCs can sense IFN-Is released by tumor cells through tumor DNA-mediated activation of cGAS-STING pathway (Chen et al, Nat Immunol, 2016).

11. Figure 6E; I don't understand the STING knockout data. STING deletion does reduce IFN expression, but then why is the effect of STING deletion in CDK6 deleted cells not the same as CDK6 deleted cells in IFNAR1 deleted mice?

Response: Yes, the effect of IFNAR1 knockout in mice seems a little stronger than STING deletion in CDK6 deleted cells. The reason may be that IFNAR knockout in mice completely cut off the type I interferon signaling pathway in host, whereas STING knockout in CDK6 deleted cells obviously weakens but still retain some effect of type I interferon signaling. In fact, these two tumor growth curves were similar, smaller than WT cells growing in WT mice but larger than *Cdk6*^{-/-} cells growing in WT mice.

12. Figure 7E; This is a very different effect. Palbociclib will inhibit proliferation and may start to trigger senescence in PBMCs. Part of the senescence program is SASP expression which can include ISGs. As you have presented no evidence that CDK4 or CDK6 deletion promotes senescence in your models this is a quite different effect.

Response: We have tested the SASP levels of *Cdk4*^{-/-} and *Cdk6*^{-/-} cells according to your suggestion. It is interesting to find that unlike *Cdk4*^{-/-} cells, *Cdk6*^{-/-} cells are becoming senescent, which may explain why *Cdk6*^{-/-} cells proliferate more slowly (Supplementary Figures 3H and I). We cannot exclude the possibility that the induced ISGs could be partially facilitated by SASP response in *Cdk6*^{-/-} cells. However, double knockout of STING and CDK6 down-regulated the ISGs induced by *Cdk6* single knockout (Figure 6D), which exerted the important role of cGAS-STING signaling in *Cdk6*-deficiency induced type I interferon responses. The scientific question you raised is meaningful, we would like to study the role of SASP induced by *Cdk6* knockout in our future study.

Response to Reviewer #3

1. It is well known that CDK4/CDK6 blockade triggers a robust anti-tumor immune response, which the authors cite. The authors suggest that they observed activation of cGAS-STING signaling because of *Cdk4* KO or *Cdk6* KO, and not MAVS-dependent signaling as observed by Goel et al because abemaciclib also inhibits CDK2. Does inhibition of CDK2 (for example, treatment with abemaciclib or shRNA to *Cdk2*) in *Cdk4* KO or *Cdk6* KO cells activate MAVS-dependent signaling? Does treatment with Palbociclib which has been reported to act via cGAS-STING signaling (Liu et al. *Front. Immunol* 2021) of *Cdk4* KO or CDK6 KO cells have an additional effect?

Response: Thank you for your constructive comments on my manuscript. Our previous work showed that MAVS and CDK2 double knockout could reverse the anti-tumor effect of CDK2 single knockout, which proved that CDK2 deficiency induced anti-tumor immunity was dependent on MAVS signaling (Chen et al, *Cancer Immunol Res*, 2022). We did not study whether inhibition of CDK2 in CDK4 or CDK6 KO cells could activate MAVS-dependent signaling because double knockout of MAVS and CDK4/CDK6 did not affect CDK4/6 deficiency-induced anti-tumor effect, which ruled out the possibility that MAVS take part in CDK4 or CDK6 deletion induced anti-tumor effect.

We have conducted relevant experiments to answer your second question. Interestingly, palbociclib seems to have an additional effect on *Cdk4*^{-/-} cells but not *Cdk6*^{-/-} cells. This is an interesting finding, we would like to search the reason for this difference in our future study.

2. CDK4/CDK6 blockade is currently only approved in breast cancer. The majority of the data is from MCA205 fibrosarcoma and TC1 lung. It would be important to show that CDK4/CDK6 blockade in breast tumor models activates the cGAS-STING pathway leading to immune infiltration.

Response: Thank you for the constructive suggestion, we agree that this experiment will strengthen our conclusion. We treat breast cancer cell line 4T1 with palbociclib (0.5uM) for 10 days and found that palbociclib could activate the cGAS-STING pathway and subsequent type I interferon pathway in both MCA205 and 4T1 cell lines (added in Figures 7 B and C).

3. In Figure 2I, the authors show that CD8⁺ T-cell depletion abrogates the phenotype of Cdk4 KO in mice. Does this also occur for Cdk6 KO?

Response: In vivo study showed that tumors grew to similar sizes in nude mice injected with WT or *Cdk4*^{-/-} MCA205 and TC-1 cells (Figures 1F and G), suggesting the anti-tumor effect of *Cdk4* knockout was immune-dependent. So we used anti-CD8 antibody to deplete CD8⁺ T cells to verify the important role of CD8⁺ T cells in CDK4-deficiency induced anti-tumor effect. However, *Cdk6* deleted cells still grew much smaller and slower than WT cells in both nude mice and NSG mice (Figures 5B and C). The factors contributing to *Cdk6*-deficiency induced anti-tumor effect are more complex, including cell growth inhibition and immune response. That is the reason why we did not use anti-CD8 antibody in *Cdk6* knockout *in vivo* study.

4. Figure 2A and Figure 3A are very similar. Similarly, Figure 2B and 3B are very similar.

Response: Figure 2A and 2B are RNA-seq analysis of WT and *Cdk4*^{-/-} MCA205 graft tumor tissues and Figure 3A and 3B are RNA-seq analysis of WT and *Cdk4*^{-/-} MCA205 cell lines. So there may exist some similar signaling pathways.

5. Figure 6D and E figure legends are flipped.

Response: We feel very sorry to make this mistake and we have corrected this in the manuscript. Thank you a lot for pointing this out.

Reviewers' comments:

Reviewer #1 (Remarks to the Author):

The authors have described a novel mechanism that relies on cGAS-STING pathway to facilitate an immune response that halts tumour cell growth following inhibition of the CDK4/6 pathway.

The authors have addressed all of my concerns which has strengthened their manuscript and I have no further comments. I support publication of this study

Reviewer #2 (Remarks to the Author):

Thank you for your detailed responses. They have assisted in understanding the data. You have now found that RB in MCA205 cells is highly mutated, and your Supp Fig 1 shows that palbociclib has a very modest effect on MCA205 proliferation which only becomes significant after 72 h treatment, the FACS data showing that there is only modest change in S and G2/M phase population with even extended palbo treatment all indicate that MCA205 have a defective RB. The effects of CDK4/6 inhibitors on RB functional cell lines is far more robust and rapid. The addition of TC1 cell line data only confirms this as TC1 express HPV E7 that knocks out RB and is very effective in bypassing CDK4/6 inhibitor induced G1 phase cell cycle arrest and senescence. This does not alter your data. Indeed, it makes it far more interesting as it indicates that CDK4/6 inhibitors may be very useful in RB defective tumours, although this would be through the very different mechanism you define in this manuscript, rather than the senescence-associated mechanism proposed for RB functional tumours. In essence, your original concept of investigating the real mechanism of inhibition of CDK4/6 in cancer therapy has not been realised, as in cancer treatment only RB functional tumours receive CDK4/6 inhibitors and your CDK4/6 knockout models are in RB defective tumours. However, the large body of convincing data you present shows that these inhibitor may also be useful in RB-defective tumours. The difference in the effects of CDK4 and CDK6 knockout is also interesting and novel. It will be necessary to rephase this manuscript to reflect this very different model you are testing here, but the data and their implications are very important.

Reviewer #3 (Remarks to the Author):

This revised study by Huimin Fan et al. provides additional insights into how CDK4 and CDK6 blockade, using genetically engineered Cdk4 and Cdk6 KO mouse models, activates cGAS-STING signaling, induces type I interferon gene expression, leading to an innate and adaptive immune response and antitumor immunity. They have now extensively interrogated activation of cGAS-STING

signaling, including in breast cancer models as requested given CDK4/6 inhibitors are only currently indicated in this disease. The authors have addressed my concerns.

Dear Editor Valeria Naim,

This is a response letter for manuscript #COMMSBIO-23-1235-T. Thank you for your and all the reviewers' comments. We are glad that our revision generally fulfilled the reviewers' requirements. We have revised our manuscript, with the discussion that our study applies to both tumors with functional RB and tumors with defective RB. The specific reasons are listed in the response to Reviewer 2#. We added relative content in Result and Discussion to make it clear, which was highlighted in the revised manuscript. We would like to study the effect of Rb status on the effect of CDK4/6 inhibitors in future research. Given the available evidence and clinical guidance significance of this article, we choose not to emphasize that our study only applies to RB-detective tumors. We sincerely hope to get your understanding.

Best regards,

Yang Heng, PHD

Suzhou Institute of Systems Medicine, Institute of Systems Medicine, Chinese Academy of Medical Sciences & Peking Union Medical College

The following is our point-by-point reply to reviewer's comments.

Reviewer 2#: Thank you for your detailed responses. They have assisted in understanding the data. You have now found that RB in MCA205 cells is highly mutated, and your Supp Fig 1 shows that palbociclib has a very modest effect on MCA205 proliferation which only becomes significant after 72 h treatment, the FACS data showing that there is only modest change in S and G2/M phase population with even extended palbo treatment all indicate that MCA205 have a defective RB. The effects of CDK4/6 inhibitors on RB functional cell lines is far more robust and rapid. The addition of TC1 cell line data only confirms this as TC1 express HPV E7 that knocks out RB and is very effective in bypassing CDK4/6 inhibitor induced G1 phase cell cycle arrest and senescence. This does not alter your data. Indeed, it makes it far more interesting as it indicates that CDK4/6 inhibitors may be very useful in RB defective tumours, although this would be through the very different mechanism you define in this manuscript, rather than the senescence-associated mechanism proposed for RB functional tumours. In essence, your original concept of investigating the real mechanism of inhibition of CDK4/6 in cancer therapy has not been realised, as in cancer treatment only RB functional tumours receive CDK4/6 inhibitors and your CDK4/6 knockout models are in RB defective tumours. However, the large body of convincing data you present shows that these inhibitor may also be useful in RB-defective tumours. The difference in the effects of CDK4 and CDK6 knockout is also interesting and novel. It will be necessary to rephrase this manuscript to reflect this very different model you are testing here, but the data and their implications are very important.

Response: Many thanks for your discussion in depth with us. We did not pay much attention in Rb status in our previous work so thank you for pointing this out. We have read some relative research in this field. In one study, to address RB1 functionality, the author examined the effect of CDK4/6 inhibitor on RB1 phosphorylation (Willey et al, *Mol Cancer Ther.* 2023). We observed a decrease of RB1 phosphorylation in *Cdk4* and *Cdk6* knockout MCA205 cells compared with WT cells (Supplementary Figure 3E). In our separate work, knockout of *Cdk2* in MCA205 could also

reduce the expression of p-RB (Chen Yu, *Cancer Immunol Res*, 2022), demonstrating that RB1 is functional in MCA205 cells. In Wildey's study, they also conducted cell cycle analysis after palbociclib treatment by FACS, it is shown that different cells exert different sensitivity to palbociclib treatment, despite the fact that they are all RB1^{WT} cells. It is possible that the sensitivity of palbociclib treatment could be influenced by cell types and drug concentration. In our study, there is a significant difference ($**p < 0.01$) of G1 cell percentage after palbociclib treatment in MCA205 cells (Supplementary Figure 3E). In addition, Wildey, et al found that growth inhibition by palbociclib was dependent on the presence of RB1 protein so they draw a conclusion that "if RB1 is expressed, it is likely to exist in a functional and targetable E2F pathway". We have also proved that RB1 is expressed in MCA205 cells (Supplementary Figure 3E) so it seems that RB in MCA205 is functional. TC1 cells were obtained from C57BL/6 primary mouse lung cells transfected with HPV E6/E7 oncogenes. As you pointed out, E7 interacts with and inactivates RB, leading to uncontrolled cell proliferation. The functional loss of Rb may affect the sensitivity of CDK4/6 inhibitors treatment, but it seems not affect tumor immunogenicity induced by CDK4/6i. So our study may apply to both tumors with functional RB and tumors with defective RB. We have added relative content in our discussion part (page 24, line 505-511) . We would like to study the interesting effect of RB status (expression, mutation, functional restriction) on CDK4/6 inhibitors treatment in our future study, which needs much more work. Thank you again for pointing this out to make our research to have a boarder range of applications.